# TUNE++: Topology-Guided Uncertainty Estimation for Reliable 3D Medical Image Segmentation

**Ashim Dhor**[1]                        ASHIMDHOR2003@GMAIL.COM
**Abhirup Banerjee**[2]            ABHIRUP.BANERJEE@ENG.OX.AC.UK
**Tanmay Basu**[1]                      TANMAY@IISERB.AC.IN
[1] *Indian Institute of Science Education and Research Bhopal, India*
[2] *Institute of Biomedical Engineering, Department of Engineering Science, University of Oxford, UK*

**Editors:** Accepted for publication at MIDL 2026

## Abstract

Deep learning models for medical image segmentation lack mechanisms to assess their own reliability, leading to two critical failures: they provide no uncertainty estimates to distinguish confident predictions from error-prone ones, and often produce anatomically implausible segmentations or incorrect connectivity that violate known structural constraints. We observe that uncertainty and topology are intrinsically linked and anatomically complex regions naturally exhibit higher prediction uncertainty, while uncertain predictions require stronger enforcement of structural constraints. Building on this insight, we propose TUNE++, a unified framework that jointly learns segmentation, uncertainty quantification, and topology preservation through a novel Topology-Uncertainty aware Paired Attention (TUPA) mechanism. Our method decomposes uncertainty into aleatoric and epistemic components while simultaneously enforcing anatomical correctness through persistent homology-based constraints. A key innovation is our topology-uncertainty alignment loss that minimizes the discrepancy between predicted total uncertainty and a topological complexity score computed from organ boundaries, multi-organ junction counts, and critical points extracted from persistence diagrams, teaching the model to be uncertain precisely where anatomical structure is geometrically complex. Our empirical results demonstrate that joint modeling of TUNE++ produced enhanced segmentation accuracy, well-calibrated uncertainty estimates that successfully identify errors, substantial reduction in topological violations, and learned confidence that correlates strongly with anatomical complexity. Our source code will be available at: https://github.com/AshimDhor/tune_plus_plus.

**Keywords:** Medical image segmentation, Uncertainty quantification, Topology, Homology.

## 1. Introduction

Deep learning models, including recent transformer-based architectures (Cao et al., 2022; Hatamizadeh et al., 2022; Tang et al., 2022), achieve strong segmentation accuracy but remain difficult to deploy clinically due to limited reliability assessment. Without uncertainty quantification, clinicians cannot distinguish high-confidence predictions from error-prone cases, while models frequently generate anatomically implausible segmentations – organs with holes or disconnected components – violating known structural constraints. Recent work has separately addressed these issues. Uncertainty quantification methods (Roy et al., 2019; Wang et al., 2019; Jungo and Reyes, 2019) estimate prediction confidence but ignore anatomical structures. Topology-preserving approaches (Hu et al., 2019; Shit et al., 2021; Clough et al., 2020) enforce structural correctness but provide no uncertainty estimates.

We observe that these aspects are fundamentally interconnected: topologically complex regions are inherently more difficult to segment and should exhibit elevated uncertainty, while uncertain predictions require stronger enforcement of topological constraints. A detailed literature review is provided in Appendix A.1

We introduce **TUNE++** (**T**opology and **UN**certainty-aware **E**fficient transformers), a unified framework that jointly learns segmentation, uncertainty quantification, and topology preservation. Our contribution is the **T**opology-**U**ncertainty aware **P**aired **A**ttention (TUPA) block, which extends efficient paired attention (EPA) (Shaker et al., 2024) with two innovations: a topology-aware attention branch that focuses on anatomically critical regions identified through persistent homology, and an uncertainty-guided adaptive fusion mechanism that dynamically weights spatial, channel, and topological features based on prediction confidence. We introduce a topology-uncertainty alignment loss that enforces correlation between uncertainty estimates and topological complexity, ensuring the model exhibits higher uncertainty at boundaries, junctions, and regions with complex anatomical structure. We evaluate TUNE++ on three benchmarks spanning different anatomical regions (Synapse, ACDC, BTCV), achieving state-of-the-art segmentation accuracy (mean DSC 89.4%), topological error reduction (72%, Betti 1.94→0.54), and superior uncertainty calibration (ECE 0.043), demonstrating that joint topology-uncertainty modeling produces synergistic improvements over approaches addressing these objectives in isolation.

## 2. Methods

Medical image segmentation faces two linked challenges: models produce anatomically implausible outputs (disconnected organs, spurious holes) because they lack structural constraints, and they provide no confidence estimates to distinguish reliable from error-prone predictions. We observe that anatomically complex regions-boundaries, junctions, irregular structures-are precisely where models should express higher uncertainty, while uncertain predictions should trigger stronger structural enforcement. We propose TUNE++ (**T**opology and **UN**certainty-aware **E**fficient transformers), a unified framework that jointly learns segmentation, uncertainty quantification, and topology preservation by explicitly coupling these objectives. Given a 3D medical image $\mathbf{x} \in \mathbb{R}^{H \times W \times D}$ (height $H$, width $W$, depth $D$), our model learns a mapping that produces three outputs: (1) segmentation mask $\mathbf{y} \in \{0, 1\}^{H \times W \times D \times C}$ for $C$ anatomical classes, (2) uncertainty estimates decomposed into aleatoric uncertainty $\boldsymbol{\sigma}_a^2 \in \mathbb{R}_+^{H \times W \times D \times C}$ (data uncertainty) and epistemic uncertainty $\boldsymbol{\sigma}_e^2 \in \mathbb{R}_+^{H \times W \times D \times C}$ (model uncertainty), and (3) topological descriptors encoding structural correctness through persistence diagrams. We formalize this as:

$$\theta^* = \arg\min_\theta \mathbb{E}_{(\mathbf{x}, \mathbf{y}^*)} \left[ \mathcal{L}_{\text{seg}} + \lambda_1 \mathcal{L}_{\text{topo}} + \lambda_2 \mathcal{L}_{\text{unc}} + \lambda_3 \mathcal{L}_{\text{calib}} + \lambda_4 \mathcal{L}_{\text{hier}} \right] \tag{1}$$

where $\mathcal{L}_{\text{seg}}$ ensures accurate segmentation, $\mathcal{L}_{\text{topo}}$ enforces anatomical structure, $\mathcal{L}_{\text{unc}}$ learns uncertainty decomposition, $\mathcal{L}_{\text{calib}}$ ensures well-calibrated confidence, and $\mathcal{L}_{\text{hier}}$ maintains multi-scale consistency. Our key innovation is the explicit coupling between topology and uncertainty through the TUPA attention mechanism, detailed in Section 2.2.

### 2.1. Architecture Overview

TUNE++ adopts a four-stage hierarchical architecture operating at spatial resolutions $\{H/4, H/8, H/16, H/32\}$ with progressively increasing channel dimensions $\{C_1, 2C_1, 4C_1, 8C_1\}$ where $C_1 = 32$ is the base channel count. This multi-scale design serves two purposes: coarse resolutions (H/32, H/16) capture global organ-level context and inter-organ relationships, while fine resolutions (H/8, H/4) preserve boundary details critical for accurate delineation. Figure 1 illustrates the complete architecture.

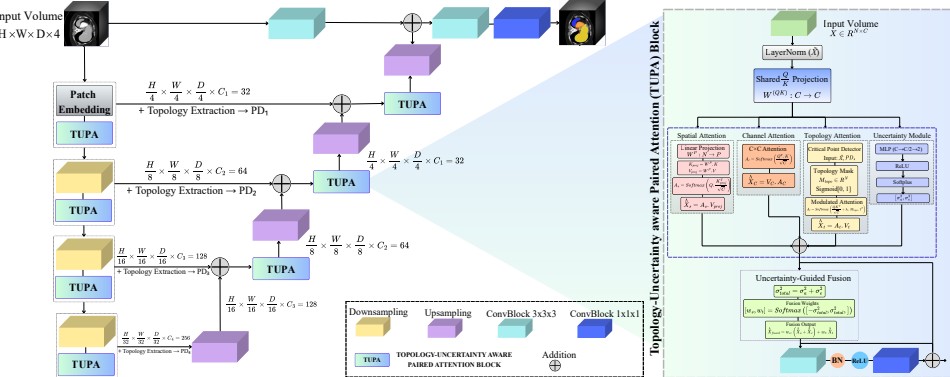

Figure 1: TUNE++ architecture. The encoder extracts multi-scale features while computing topological descriptors (persistence diagrams) at each stage. TUPA blocks integrate spatial, channel, and topology-aware attention. The decoder produces segmentation and uncertainty estimates through specialized heads.

#### 2.1.1. Input Processing and Patch Embedding

Following Vision Transformer (ViT) design (Dosovitskiy et al., 2021) , we divide the input volume into non-overlapping 3D patches of size $(P_h, P_w, P_d) = (4, 4, 2)$. The asymmetric patch dimensions accommodate typical medical image anisotropy where through-plane (axial) resolution is coarser than in-plane resolution. This yields $N = \frac{HWD}{32}$ tokens, each linearly projected to $C_1 = 32$ dimensions with learnable positional embeddings encoding spatial structure. This tokenization converts the volumetric input into a sequence suitable for transformer processing while preserving 3D spatial relationships.

#### 2.1.2. Encoder Architecture

Each of the four encoder stages contains three components working together to extract increasingly abstract features while tracking their topological properties. First, stride-2 convolution with $3 \times 3 \times 3$ kernels performs downsampling, reducing spatial dimensions by half while doubling channel count to progressively capture higher-level features. Second, our novel TUPA block (detailed in Section 2.2) processes these features through three parallel branches: spatial attention, channel attention, and topology-aware attention, and then fuses them using uncertainty-guided weighting, allowing the model to adaptively balance data-driven and structure-driven information. Third, at each encoder stage $s$, we extract

topological features from the feature maps using persistent homology that analyzes multi-scale structure by treating feature activations as height functions and tracking when topological features like connected components, holes, and voids appear and disappear across thresholds. This produces a persistence diagram $\text{PD}_s = \{(b_i, d_i)\}$, where each (birth, death) coordinate pair represents a topological feature and its persistence $|d_i - b_i|$ quantifies how significant it is. Long-lived features with large persistence correspond to true anatomical structures, while short-lived features represent noise. This topological information then guides the TUPA attention mechanism to focus on structurally critical regions, creating a feedback loop between topology extraction and feature processing.

### 2.1.3. Decoder Architecture and Output Heads

The decoder mirrors the encoder with four stages, each containing: $2 \times 2 \times 2$ transposed convolutions with stride 2 for upsampling, doubling spatial dimensions while halving channels, skip connections from corresponding encoder stages concatenated before TUPA blocks (following UNet design (Ronneberger et al., 2015)), and TUPA blocks for feature refinement. The final decoder output at resolution $H/2 \times W/2 \times D/2$ with $C_1 = 32$ channels feeds into four parallel heads that produce our different outputs. The segmentation head applies $\text{Conv}_{3 \times 3 \times 3}$ followed by $\text{Conv}_{1 \times 1 \times 1}$, then uses bilinear upsampling to reach original resolution and applies softmax to produce $\mathbf{y} \in \mathbb{R}^{H \times W \times D \times C}$. For uncertainty quantification, we have two separate heads: the aleatoric uncertainty head uses an MLP with architecture 256 $\to 128 \to C$ followed by softplus activation to produce $\boldsymbol{\sigma}_a^2 \in \mathbb{R}_+^{H \times W \times D \times C}$, where softplus ensures the variance remains positive and captures inherent data ambiguity. The epistemic uncertainty head uses the same MLP structure to generate an initialization $\boldsymbol{\sigma}_{e,\text{init}}^2$, with the final epistemic uncertainty computed via Monte Carlo dropout at inference (detailed in Section 2.4). Finally, the topology descriptor head computes a persistence diagram on the final segmentation $\mathbf{y}$ for topology loss computation.

## 2.2. Topology-Uncertainty Aware Paired Attention (TUPA)

Standard self-attention in transformers has quadratic complexity $\mathcal{O}(N^2 C)$ where $N$ is the number of tokens and $C$ is the channel dimension, making it prohibitively expensive for 3D medical volumes. Moreover, standard attention treats all spatial locations equally, failing to account for (1) structural importance (organ boundaries and junctions require more careful modeling than homogeneous regions), and (2) prediction confidence (uncertain regions should leverage stronger structural priors). TUPA addresses both limitations by: (a) decomposing attention into efficient spatial, channel, and topology branches, reducing complexity to $\mathcal{O}(NPC)$ where $P \ll N$, and (b) dynamically fusing these branches using predicted uncertainty to adaptively balance data-driven and structure-driven features.

### 2.2.1. Shared Query-Key Projection

Given input features $\mathbf{X} \in \mathbb{R}^{N \times C}$ where $N = h \cdot w \cdot d$ is the number of spatial tokens at the current scale, we first apply layer normalization $\tilde{\mathbf{X}} = \text{LayerNorm}(\mathbf{X})$ for training stability. We then compute shared query and key projections:

$$\mathbf{Q}_{\text{shared}} = \mathbf{K}_{\text{shared}} = \mathbf{W}^{QK}\tilde{\mathbf{X}} \in \mathbb{R}^{N \times C} \tag{2}$$

where $\mathbf{W}^{QK} \in \mathbb{R}^{C \times C}$ is a learnable projection. Sharing queries and keys across attention branches reduces parameters while enabling consistent attention patterns, following efficient attention designs (Shaker et al., 2024).

### 2.2.2. Efficient Spatial and Channel Attention

To avoid quadratic $\mathcal{O}(N^2)$ complexity, we project keys and values to lower dimension $P \ll N$, reducing spatial attention complexity to $\mathcal{O}(NPC)$:

$$\mathbf{K}_{\mathrm{proj}} = (\mathbf{W}^P \mathbf{K}_{\mathrm{shared}}^T)^T \in \mathbb{R}^{P \times C}, \tag{3}$$

$$\mathbf{V}_{\mathrm{proj}}^{\mathrm{spatial}} = (\mathbf{W}^P (\mathbf{V}_{\mathrm{spatial}})^T)^T \in \mathbb{R}^{P \times C}, \tag{4}$$

$$\hat{\mathbf{X}}_{\mathrm{spatial}} = \mathrm{Softmax}\left(\frac{\mathbf{Q}_{\mathrm{shared}} \mathbf{K}_{\mathrm{proj}}^T}{\sqrt{C}}\right) \mathbf{V}_{\mathrm{proj}}^{\mathrm{spatial}}, \tag{5}$$

where $\mathbf{V}_{\mathrm{spatial}} = \mathbf{W}^{V_{\mathrm{spatial}}} \tilde{\mathbf{X}} \in \mathbb{R}^{N \times C}$ is the spatial value projection and $\mathbf{W}^P \in \mathbb{R}^{P \times N}$ performs learned dimensionality reduction. In parallel, channel attention models feature interdependencies by attending across the channel dimension rather than spatial locations, computing a channel-wise attention matrix that recalibrates feature importance:

$$\mathbf{A}_c = \mathrm{Softmax}\left(\frac{\mathbf{Q}_{\mathrm{shared}}^T \mathbf{K}_{\mathrm{shared}}}{\sqrt{C}}\right) \in \mathbb{R}^{C \times C}, \tag{6}$$

$$\hat{\mathbf{X}}_c = \mathbf{V}_c \mathbf{A}_c, \tag{7}$$

where $\mathbf{V}_c = \mathbf{W}^{V_c} \tilde{\mathbf{X}}$ is the channel value projection. These two branches capture both spatial relationships between locations and semantic relationships between feature channels, providing complementary representations that are later fused based on predicted uncertainty.

### 2.2.3. Topology-Aware Attention Branch

The key innovation of TUPA is incorporating structural knowledge through topological attention. We identify anatomically critical regions using a Critical Point Detector:

$$\mathbf{M}_{\mathrm{topo}} = \mathrm{CriticalPointDetector}(\tilde{\mathbf{X}}, \mathrm{PD}_s) \in \mathbb{R}^N \tag{8}$$

where $\mathrm{PD}_s$ is the persistence diagram extracted at encoder stage $s$. The detector (implemented as a 3-layer CNN) takes as input both the current features $\hat{\mathbf{X}}$ and the topological descriptor $\mathrm{PD}_s$, producing a scalar importance score for each spatial location. High scores are assigned to organ boundaries (transitions between tissues), multi-organ junctions, and critical points identified from the persistence diagram (locations where topological features appear or disappear in the filtration). We then compute topology-modulated attention:

$$\mathbf{A}_t = \mathrm{Softmax}\left(\frac{\mathbf{Q}_{\mathrm{shared}} \mathbf{K}_{\mathrm{shared}}^T}{\sqrt{C}} + \lambda_t \mathbf{M}_{\mathrm{topo}} \mathbf{1}^T\right) \in \mathbb{R}^{N \times N}, \tag{9}$$

$$\hat{\mathbf{X}}_t = \mathbf{A}_t \mathbf{V}_t, \tag{10}$$

where $\mathbf{V}_t = \mathbf{W}^{V_t} \tilde{\mathbf{X}} \in \mathbb{R}^{N \times C}$ is the topology value projection, and $\lambda_t = 0.3$ controls how strongly topological structure biases attention (determined via grid search over $\{0.1, 0.2, 0.3, 0.4, 0.5\}$ on validation data). The broadcast operation $\mathbf{M}_{\mathrm{topo}} \mathbf{1}^T$ upweights attention to structurally critical regions identified by persistent homology.

2.2.4. Uncertainty-Guided Adaptive Fusion

Parallel to the three attention branches, we estimate voxel-wise uncertainty to guide how these branches should be combined:

$$[\boldsymbol{\sigma}_a^2, \boldsymbol{\sigma}_{e,\text{init}}^2] = \text{MLP}(\tilde{\mathbf{X}}) \in \mathbb{R}^{N \times 2} \tag{11}$$

where $\sigma_a^2$ captures aleatoric uncertainty and $\sigma_{e,\text{init}}^2$ initializes epistemic uncertainty. We compute total uncertainty as $\boldsymbol{\sigma}_{\text{total}}^2 = \boldsymbol{\sigma}_a^2 + \boldsymbol{\sigma}_{e,\text{init}}^2$. The key insight is that uncertain regions should rely more heavily on topological structure, while confident regions can rely on data-driven spatial/channel attention. We implement this through adaptive fusion weights:

$$[w_s, w_t] = \text{Softmax}([-\boldsymbol{\sigma}_{\text{total}}^2, \boldsymbol{\sigma}_{\text{total}}^2]) \in \mathbb{R}^{N \times 2} \tag{12}$$

where $w_s, w_t \in \mathbb{R}^N$ are per-voxel weights for spatial/channel versus topology attention. When uncertainty is low ($\sigma_{\text{total}}^2 \approx 0$), the softmax produces balanced weights $w_s \approx w_t \approx 0.5$, allowing data-driven features to dominate. When uncertainty is high ($\sigma_{\text{total}}^2 \gg 0$), topology weight increases ($w_t \to 1, w_s \to 0$), enforcing structural constraints where the model is uncertain. The three attention outputs are fused as:

$$\hat{\mathbf{X}}_{\text{fused}} = w_s \odot (\hat{\mathbf{X}}_s + \hat{\mathbf{X}}_c) + w_t \odot \hat{\mathbf{X}}_t \tag{13}$$

where $\odot$ denotes element-wise multiplication. Finally, convolutional refinement processes fused features:

$$\mathbf{X}_{\text{out}} = \text{Conv}_{1 \times 1 \times 1}(\text{BN}(\text{ReLU}(\text{Conv}_{3 \times 3 \times 3}(\hat{\mathbf{X}}_{\text{fused}})))). \tag{14}$$

This completes the TUPA block, which is applied at each encoder and decoder stage, progressively refining features with topology-uncertainty aware attention.

## 2.3. Training Objectives

Training TUNE++ requires balancing five complementary objectives, each targeting a distinct aspect of reliable segmentation. The total training objective is:

$$\mathcal{L}_{\text{total}} = \mathcal{L}_{\text{seg}} + \lambda_1 \mathcal{L}_{\text{topo}} + \lambda_2 \mathcal{L}_{\text{unc}} + \lambda_3 \mathcal{L}_{\text{calib}} + \lambda_4 \mathcal{L}_{\text{hier}} \tag{15}$$

$\mathcal{L}_{\text{seg}}$ ensures segmentation accuracy (unweighted, coefficient 1.0), $\mathcal{L}_{\text{topo}}$ enforces topological correctness ($\lambda_1 = 0.3$), $\mathcal{L}_{\text{unc}}$ learns uncertainty decomposition and calibration ($\lambda_2 = 0.2$), $\mathcal{L}_{\text{calib}}$ ensures confidence calibration ($\lambda_3 = 0.1$, serving as a regularizer), and $\mathcal{L}_{\text{hier}}$ maintains multi-scale topology consistency ($\lambda_4 = 0.15$, also a regularizer). This weight hierarchy reflects our task priorities: topology provides the strongest structural constraints preventing anatomically impossible outputs, uncertainty enables reliability assessment critical for clinical deployment, and calibration together with hierarchical consistency serve as a regularizer refining model behavior. Detailed analysis is given in Appendix A.5.

**Segmentation Loss** combines Dice and cross-entropy to handle class imbalance while providing dense gradients. Complete formulations are provided in Appendix A.3.1.

$$\mathcal{L}_{\text{seg}} = \mathcal{L}_{\text{Dice}} + \mathcal{L}_{\text{CE}} \tag{16}$$

**Topology Preservation Loss** ensures anatomically plausible structures by enforcing correct connectivity, preventing spurious holes, and avoiding fragmented organs. We combine three complementary topological constraints:

$$\mathcal{L}_{\text{topo}} = \mathcal{L}_{\text{PH}} + 0.5\mathcal{L}_{\text{Betti}} + 0.3\mathcal{L}_{\text{critical}}. \tag{17}$$

where $\mathcal{L}_{\text{PH}}$ measures multi-scale structural similarity through persistent homology (capturing when topological features appear and disappear across scales), $\mathcal{L}_{\text{Betti}}$ penalizes incorrect topological invariants, and $\mathcal{L}_{\text{critical}}$ ensures critical topological locations are correctly positioned. Complete formulations are provided in Appendix A.3.2.

**Uncertainty Loss:** We decompose total uncertainty into aleatoric and epistemic components, then align them with topological complexity:

$$\mathcal{L}_{\text{unc}} = \mathcal{L}_{\text{aleatoric}} + \mathcal{L}_{\text{epistemic}} + 0.5\mathcal{L}_{\text{align}}. \tag{18}$$

Aleatoric Uncertainty Loss learns heteroscedastic noise following (Kendall and Gal, 2017):

$$\mathcal{L}_{\text{aleatoric}} = \frac{1}{N_{\text{vox}}C} \sum_{i=1}^{N_{\text{vox}}} \sum_{c=1}^{C} \left( \frac{\|p_{i,c} - g_{i,c}\|^2}{2\sigma_{a,i,c}^2} + \log(\sigma_{a,i,c}^2 + \epsilon) \right). \tag{19}$$

The first term penalizes prediction error normalized by predicted noise, while the second term prevents the model from trivially minimizing loss by predicting infinite uncertainty. Epistemic Uncertainty Loss enforces consistency across multiple stochastic forward passes:

$$\mathcal{L}_{\text{epistemic}} = \text{KL}(p_{\text{single}}\|p_{\text{MC}}) = \frac{1}{N_{\text{vox}}C} \sum_{i,c} p_{i,c}^{\text{single}} \log \frac{p_{i,c}^{\text{single}}}{p_{i,c}^{\text{MC}}} \tag{20}$$

where $p_{\text{single}}$ is a single forward pass prediction, and $p_{\text{MC}}$ is the mean over dropout samples. This KL divergence encourages the learned epistemic initialization to approximate the true variability captured by MC dropout. Topology-Uncertainty Alignment Loss creates synergy by correlating prediction uncertainty with topological complexity:

$$\mathcal{L}_{\text{align}} = \frac{1}{N_{\text{vox}}} \sum_{i=1}^{N_{\text{vox}}} \|\sigma_{\text{total},i}^2 - C_{\text{topo},i}\|_2^2 \tag{21}$$

where the topological complexity score combines three components:

$$C_{\text{topo},i} = w_b B_i + w_j J_i + w_a A_i \tag{22}$$

with $B_i \in \{0,1\}$ indicating boundaries (organ edges), $J_i \in \mathbb{Z}_+$ counting organ junctions (where 3+ structures meet), and $A_i \in [0,1]$ measuring topological anomalies from persistence diagrams. The weights $w_b = 1.0$, $w_j = 2.0$, $w_a = 3.0$ reflect increasing topological severity: boundaries are baseline features, junctions involve multiple organs requiring stronger enforcement, and anomalies represent severe violations warranting highest penalty.

**Calibration Loss** ensures predicted confidence matches actual correctness probability through Expected Calibration Error (ECE) and Brier Score (BS):

$$\mathcal{L}_{\text{calib}} = \text{ECE} + 0.5 \cdot \text{Brier} = \sum_{m=1}^{M} \frac{|B_m|}{N_{\text{vox}}} |\text{acc}(B_m) - \text{conf}(B_m)| + \frac{0.5}{N_{\text{vox}}C} \sum_{i,c} (p_{i,c} - g_{i,c})^2 \tag{23}$$

where predictions are discretized into $M = 10$ confidence bins $B_m$, $\text{acc}(B_m)$ is the empirical accuracy within bin $m$, and $\text{conf}(B_m)$ is the average predicted confidence.

**Hierarchical Topology Consistency Loss:** We expect topology to remain stable across scales-downsampling shouldn't fundamentally change whether an organ appears as one piece or fragments:

$$\mathcal{L}_{\text{hier}} = \sum_{s=2}^{4} W_2(\text{PD}_s, \text{Downsample}(\text{PD}_{s-1})) \tag{24}$$

where $\text{PD}_s$ is the persistence diagram at encoder stage $s$, and $\text{Downsample}(\cdot)$ recomputes the diagram at the next coarser scale, preventing scale-dependent artifacts.

## 2.4. Inference Protocol

At inference, we perform $K = 25$ stochastic forward passes with dropout enabled (Monte Carlo dropout (Gal and Ghahramani, 2016)), each producing prediction $\mathbf{y}_k$ and aleatoric estimate $\boldsymbol{\sigma}_{a,k}^2$. The final prediction is the mean output $\mathbf{y}_{\text{final}} = \frac{1}{K} \sum_{k=1}^{K} \mathbf{y}_k$. Aleatoric uncertainty is estimated as $\boldsymbol{\sigma}_a^2 = \frac{1}{K} \sum_{k=1}^{K} \boldsymbol{\sigma}_{a,k}^2$, epistemic uncertainty is computed from predictive variance $\boldsymbol{\sigma}_e^2 = \frac{1}{K} \sum_{k=1}^{K} (\mathbf{y}_k - \mathbf{y}_{\text{final}})^2$, and total uncertainty combines both: $\boldsymbol{\sigma}_{\text{total}}^2 = \boldsymbol{\sigma}_a^2 + \boldsymbol{\sigma}_e^2$. This decomposition allows distinguishing irreducible data ambiguity from model confidence, critical for clinical decision-making.

## 3. Experiments and Results

### 3.1. Datasets and Implementation

We evaluated TUNE++ on three public benchmark datasets: Synapse multiorgan segmentation CT scans (Landman et al., 2015) containing 30 volumetric scans containing 8 abdominal organs with an 18/12 train/test split following (Zhou et al., 2023), Automatic Cardiac Diagnosis Challenge (ACDC) (Bernard et al., 2018) (100 subjects, 70/10/20 - train/validation/test split) follwoing (Zhou et al., 2023), and BTCV abdominal CT (Landman et al., 2015) (50 subjects, standard 30/20 - train/test split). Implementation details including training hyperparameters and augmentation strategies are provided in Appendix A.2. Experiments are conducted on a NVIDIA H100 (95GB) GPU. We evaluate performance across three metric groups: segmentation accuracy - Dice Similarity Coefficient (DSC), Normalized Surface Distance (NSD), 95th percentile Hausdorff Distance (HD95), uncertainty quality (ECE and Brier Score) and topological correctness (Betti Error). We introduce **Topology-Aware Uncertainty Score** (TAUS), measuring correlation between predicted uncertainty and topological complexity:

$$\text{TAUS} = \text{Pearson}(\sigma_{\text{total}}^2, C_{\text{topo}}) \tag{25}$$

where $C_{\text{topo}}$ is topological complexity score. TAUS $\in$ [-1, 1], with higher values indicating better uncertainty-anatomy alignment. Full metric definitions are given in Appendix A.4.

### 3.2. Results

Table 1 summarizes the Synapse multi-organ benchmark results, where TUNE++ achieves 89.3% mean DSC, establishing a new state-of-the-art with statistically significant gains over all baselines ($p < 0.001$). Beyond average accuracy, it offers substantially improved boundary quality, evidenced by an 89.1% NSD and HD95 reduced from 7.5mm (UNETR++) to 6.2mm. Performance gains are notable in challenging organs such as pancreas (84.2% DSC) and gallbladder (73.8%), while accuracy remains high for large organs (e.g., spleen 96.5%), demonstrating robustness across scale and anatomy. A comparison with baseline models and qualitative visual results are shown in Figure 2.

Table 1: Synapse dataset comparison. Best results in **bold**, second best underlined.

| Method | Spl | RK | LK | Gal | Liv | Sto | Aor | Pan | Mean DSC↑ | NSD↑ (%) | HD95↓ (mm) | Betti Err↓ |
|---|---|---|---|---|---|---|---|---|---|---|---|---|
| U-Net | 86.7 | 68.6 | 77.8 | 69.7 | 93.4 | 75.6 | 89.1 | 54.0 | 76.9 | 83.2 | 39.7 | 2.87 |
| nnUNet | 90.5 | 86.2 | 86.6 | 70.2 | 96.8 | 86.8 | 92.0 | 83.4 | 86.6 | 84.5 | 10.6 | 1.89 |
| UNETR | 86.7 | 85.6 | 85.6 | 56.3 | 94.6 | 70.5 | 89.8 | 60.5 | 78.4 | 76.6 | 18.6 | 2.41 |
| Swin-UNETR | 95.4 | 86.3 | 87.0 | 66.5 | 95.7 | 77.0 | 91.1 | 68.8 | 83.5 | 80.9 | 10.6 | 1.76 |
| nnFormer | 90.5 | 86.6 | 86.6 | 70.2 | 96.8 | 86.8 | 92.0 | 83.4 | 86.4 | 83.8 | 10.6 | 1.52 |
| UNETR++ | 95.8 | 87.2 | 87.5 | 71.3 | 96.4 | 86.0 | 92.5 | 81.1 | 87.2 | 86.0 | 7.5 | 1.34 |
| **TUNE++** | **96.5** | **89.1** | **89.3** | **73.8** | **97.1** | **87.8** | **93.6** | **84.2** | **89.3** | **89.1** | **6.2** | **0.34** |

*Spl=Spleen, RK=Right Kidney, LK=Left Kidney, Gal=Gallbladder, Liv=Liver, Sto=Stomach, Aor=Aorta, Pan=Pancreas.*

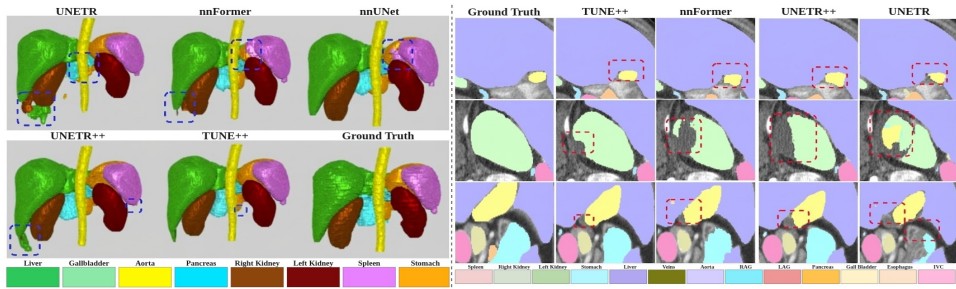

Figure 2: Qualitative results on Synapse dataset. **Left:** 3D renderings showing topological correctness. **Right:** 2D slices showing segmentation.

Table 2 presents evaluation on the ACDC dataset. TUNE++ achieves 93.8% mean DSC, establishing new state-of-the-art on this cardiac segmentation benchmark. Myocardium, the most challenging structure due to its thin walls and complex geometry, benefits from topology-aware attention. Qualitative visual results are presented in Figure 3(a).

Table 3 presents evaluation on BTCV's 13-organ segmentation task. TUNE++ achieves 84.8% mean DSC, outperforming UNETR++ baseline (82.3%, $p < 0.001$). Per-organ analysis reveals consistent improvements across all anatomical structures, with particularly notable gains on challenging small organs: right adrenal gland (+4.5%, 66.8%→71.3%), left adrenal gland (+4.7%, 68.1%→72.8%), and pancreas (+1.6%, 81.2%→82.8%). These improvements stem from topology-aware attention dynamically allocating computational

Table 2: Results on the ACDC dataset. Best results in **bold**, second-best underlined.

| Method | RV | LV | Myo | Mean DSC↑ | NSD↑ (%) | HD95↓ (mm) | Betti Err↓ |
|--------|------|------|------|-----------|----------|------------|------------|
| U-Net | 87.5 | 94.2 | 86.1 | 89.3 | 86.2 | 8.4 | 2.12 |
| nnUNet | 91.4 | 95.8 | 88.7 | 92.0 | 89.5 | 5.2 | 1.45 |
| UNETR | 88.2 | 93.6 | 84.3 | 88.7 | 85.1 | 9.8 | 2.34 |
| Swin-UNETR | 90.8 | 95.1 | 87.9 | 91.3 | 88.7 | 6.1 | 1.67 |
| nnFormer | 91.6 | 95.6 | 88.5 | 91.9 | 89.2 | 5.4 | 1.52 |
| UNETR++ | 92.1 | 96.0 | 89.2 | 92.4 | 89.8 | 5.0 | 1.38 |
| **TUNE++** | **93.8** | **96.7** | **90.9** | **93.8** | **91.5** | **4.2** | **0.42** |

*Organ abbreviations: RV = Right Ventricle, LV = Left Ventricle, Myo = Myocardium.*

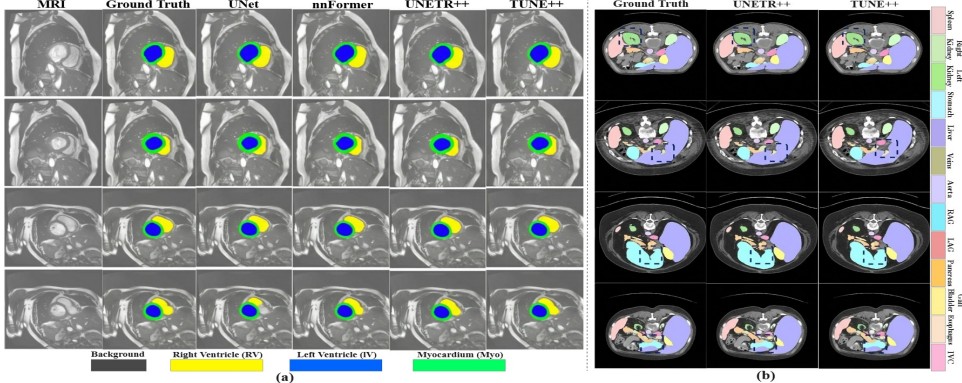

Figure 3: **(a)** Qualitative comparison on ACDC cardiac segmentation. **(b)** Multi-slice qualitative comparison on BTCV dataset.

resources to anatomically complex regions - specifically multi-organ junctions where boundaries overlap and small structures requiring precise delineation. HD95 improvement (9.8mm →8.6mm) further validates enhanced boundary precision. Visual comparisons in Figure 3(b) demonstrate superior delineation of small organs and reduced topological errors.

Table 3: Comprehensive results on BTCV 13-organ abdominal segmentation.

| Method | Spl | RK | LK | Gal | Eso | Liv | Sto | Aor | IVC | Veins | Pan | RAG | LAG | Mean DSC↑ | HD95↓ (mm) | Betti Err↓ |
|--------|------|------|------|------|------|------|------|------|------|-------|------|------|------|-----------|------------|------------|
| U-Net | 85.3 | 76.2 | 78.9 | 64.5 | 67.8 | 92.1 | 73.4 | 86.7 | 72.3 | 58.9 | 52.1 | 48.3 | 51.7 | 70.6 | 28.4 | 3.78 |
| nnUNet | 91.2 | 88.5 | 89.1 | 71.8 | 74.2 | 96.2 | 84.3 | 91.5 | 82.7 | 68.4 | 78.9 | 63.5 | 65.2 | 80.4 | 11.2 | 2.45 |
| UNETR | 87.4 | 82.3 | 84.6 | 58.9 | 69.1 | 93.8 | 76.5 | 88.2 | 76.4 | 61.2 | 64.8 | 52.3 | 54.7 | 73.9 | 18.7 | 3.12 |
| Swin-UNETR | 90.5 | 87.8 | 88.4 | 70.2 | 73.5 | 95.7 | 83.1 | 90.8 | 81.2 | 67.1 | 77.2 | 61.8 | 63.5 | 79.3 | 12.8 | 2.68 |
| nnFormer | 91.8 | 89.1 | 89.6 | 72.5 | 75.1 | 96.5 | 85.2 | 91.9 | 83.4 | 69.2 | 79.8 | 64.7 | 66.4 | 81.2 | 10.5 | 2.31 |
| UNETR++ | 92.5 | 89.7 | 90.3 | 73.4 | 76.8 | 96.8 | 86.1 | 92.3 | 84.5 | 70.9 | 81.2 | 66.8 | 68.1 | 82.3 | 9.8 | 2.12 |
| **TUNE++** | **93.8** | **91.2** | **91.7** | **74.5** | **77.4** | **97.5** | **88.3** | **93.7** | **86.9** | **73.2** | **82.8** | **71.3** | **72.8** | **84.8** | **8.6** | **1.88** |

*Spl=Spleen, RK=Right Kidney, LK=Left Kidney, Gal=Gallbladder, Eso=Esophagus, Liv=Liver, Sto=Stomach, Aor=Aorta, IVC=Inferior Vena Cava, Veins=Portal and Splenic Veins, Pan=Pancreas, RAG=Right Adrenal Gland, LAG=Left Adrenal Gland.*

Detailed uncertainty (Table 12) and topology (Table 13) evaluations are provided in Appendix A.8, confirming TUNE++'s superior calibration and topological correctness across

Table 4: Comprehensive ablation study across all datasets.

| Configuration | Synapse DSC | ACDC DSC | BTCV DSC | Mean DSC↑ | Mean Betti↓ | Mean ECE↓ | Mean ECE↓ |
|---|---|---|---|---|---|---|---|
| UNETR++ (baseline) | 87.2 | 92.4 | 82.3 | 87.3 | 1.94 | – | – |
| *Individual components:* | | | | | | | |
| + Spatial Attn only | 87.4 | 92.7 | 82.6 | 87.6 | 1.87 | – | – |
| + Channel Attn only | 87.5 | 92.6 | 82.5 | 87.5 | 1.90 | – | – |
| + Topology Attn only | 88.1 | 93.1 | 83.2 | 88.1 | 0.98 | – | – |
| + Uncertainty only | 87.3 | 92.5 | 82.4 | 87.4 | 1.89 | 0.064 | – |
| *Pairwise combinations:* | | | | | | | |
| + Spatial + Channel | 87.7 | 92.8 | 82.9 | 87.8 | 1.82 | – | – |
| + Topo + Uncertainty | 88.5 | 93.3 | 83.8 | 88.5 | 0.84 | 0.058 | 0.68 |
| *Full integration:* | | | | | | | |
| + All (fixed fusion) | 88.9 | 93.5 | 84.2 | 88.9 | 0.72 | 0.053 | 0.72 |
| *Loss ablations:* | | | | | | | |
| w/o $\mathcal{L}_{\text{align}}$ | 89.0 | 93.6 | 84.4 | 89.0 | 0.68 | 0.056 | 0.58 |
| w/o $\mathcal{L}_{\text{hier}}$ | 89.2 | 93.7 | 84.5 | 89.1 | 0.63 | 0.049 | 0.74 |
| **TUNE++ (Full)** | **89.5** | **93.8** | **84.8** | **89.4** | **0.54** | **0.043** | **0.78** |

all datasets. Table 4 presents ablation across all datasets and reveals critical insights: adding topology attention alone improves mean DSC from 87.3% to 88.1% with Betti error reduction. The Topology+Uncertainty combination (row 7) outperforms Spatial+Channel EPA (row 6), demonstrating that joint topology-uncertainty modeling provides more value than efficient paired attention alone for medical segmentation. Removing $\mathcal{L}_{\text{align}}$ causes TAUS to drop from 0.78 to 0.58 while Betti error increases from 0.54 to 0.68, confirming this loss is essential for learning uncertainty that correlates with anatomical complexity rather than generic prediction variance. Removing $\mathcal{L}_{\text{hier}}$ increases Betti error from 0.54 to 0.63, demonstrating importance of maintaining topological consistency across hierarchical scales.

## 4. Discussion

Our results validate that topology and uncertainty are complementary. Joint modeling (Table 4) exceeds individual contributions because uncertainty lacks structural priors while topology lacks enforcement strength. The alignment loss $\mathcal{L}_{\text{align}}$ teaches the network that topological complexity drives prediction difficulty. The 72% Betti reduction (1.94→0.54) exceeding topology-only improvements shows uncertainty-guided allocation prevents over-regularization while strengthening critical constraints. Superior calibration (ECE 0.043) shows topology eliminates impossible modes. TAUS correlation (r=0.78) confirms learned uncertainty reflects structural difficulty. Dataset-specific analysis demonstrates the value of joint modeling across diverse anatomical contexts. On Synapse, TUNE++ resolves spurious holes baseline methods produce (Figure 2), with consistent improvements on small structures (pancreas +3.1%, gallbladder +2.5%). On ACDC, modest DSC gains (+1.4%) accompany dramatic topological improvements (Betti 1.38→0.42, 70% reduction), revealing that standard methods achieve volumetric overlap through error averaging rather than structural

correctness - a critical distinction for anatomy-dependent clinical applications. On BTCV, gains on small organs (adrenal glands +4.5 - 4.7%) validate that topology-aware attention addresses transformers' uniform resource allocation limitations. Comprehensive topological evaluation (Table 13, Appendix A.8) demonstrates TUNE++ achieves consistent topology preservation across all datasets: mean Betti error 0.50 (72% reduction vs. UNETR++ baseline), mean topological accuracy 92.8%, validating that joint topology-uncertainty modeling produces anatomically coherent structures rather than merely voxel-accurate segmentations. TUNE++ maintains topological correctness even on complex multi-organ datasets (BTCV: 13 organs, Betti 0.58) where baselines exhibit substantial structural errors (UNETR++: 2.12), demonstrating robustness to anatomical complexity and enabling downstream clinical tasks requiring structurally valid segmentations. All improvements are statistically significant with $p < 0.001$ and large effect sizes (Cohen's $d$ 0.76–0.94; Table 16, Appendix A.11).

Loss weight selection (Appendix A.5, Figure 4) provide optimal regularization, with the hierarchy $\lambda_1 > \lambda_2 > \lambda_4 > \lambda_3$ reflecting task priorities: topology provides strongest structural constraints, uncertainty enables reliability assessment, validating our framework's design philosophy. Table 4 reveals three principles: adaptive fusion shows topology enforcement should be prediction-dependent, contradicting uniform weighting; removing $\mathcal{L}_{\mathrm{align}}$ causes TAUS collapse (0.78→0.58) and topology degradation, confirming this is a coupling mechanism; $\mathcal{L}_{\mathrm{hier}}$ prevents fine-scale errors, explaining why adding topology losses to standard networks provides limited benefit. Detailed calibration analysis (Figure 5, Appendix A.9) demonstrates that TUNE++ maintains superior probability-accuracy alignment across all confidence levels, while baseline methods exhibit systematic overconfidence at high predicted probabilities. The comparison with UNETR++ + clDice (Table 13) reveals that simply augmenting existing architectures with topology losses provides limited benefit. True topology preservation requires architecture-level integration where topological features guide attention computation and uncertainty estimates determine enforcement strength. Our TUPA block implements this through: (1) topology-aware attention branch that focuses on structurally critical regions, and (2) uncertainty-guided adaptive fusion that dynamically allocates topology enforcement based on confidence. Further, computational efficiency analysis (Table 14, Figure 6, Appendix A.10) demonstrates TUNE++ achieves pareto optimality: accuracy (89.4% DSC) with moderate computational cost (175.8G FLOPs, 68.9M parameters). Limitations and future directions are discussed in Appendix A.12.

## 5. Conclusion

TUNE++ is an unified framework for reliable 3D medical image segmentation that jointly models topology preservation and uncertainty quantification. The TUPA module establishes bidirectional reinforcement: topology guides where uncertainty should increase, while uncertainty determines where topological constraints should be enforced. By embedding topology awareness into the attention mechanism and modulating it with uncertainty, TUNE++ promotes anatomically coherent predictions while maintaining transformer efficiency. Validated across three diverse datasets (Synapse, ACDC, BTCV), TUNE++ demonstrates that topology and uncertainty are complementary rather than competing objectives, providing a principled foundation for trustworthy, clinically aligned medical AI systems.

## Acknowledgments

We thank IISER Bhopal for providing the computational resources used in this work, and the medical imaging community for making open-source datasets available. The work of Abhirup Banerjee was supported by the Royal Society University Research Fellowship under Grant URF\R1\221314.

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

## Appendix A. Appendix

### A.1. Literature Review

Vision transformers (ViT) have shown strong performance in medical imaging (Dosovitskiy et al., 2021; Chen et al., 2021). UNETR (Hatamizadeh et al., 2022), Swin-UNETR (Tang et al., 2022), and nnFormer (Zhou et al., 2023) advance transformer-based 3D segmentation through hierarchical attention and efficient volumetric designs. UNETR++ (Shaker et al., 2024) introduces EPA with linear complexity, offering an improved accuracy-efficiency trade-off. However, existing models do not incorporate uncertainty estimation or topological correctness. Uncertainty estimation enables trustworthy predictions through confidence quantification. Bayesian approaches (Gal and Ghahramani, 2016; Agarwal et al., 2024; Dhor and Bharadwaj, 2026) approximate posterior distributions through Monte Carlo dropout to estimate epistemic uncertainty, while Kendall and Gal (2017) decomposed uncertainty into aleatoric and epistemic components through learnable variance parameters. Ensemble methods (Lakshminarayanan et al., 2017) achieve uncertainty through prediction variance across independently trained models at significant computational cost. Probabilistic segmentation methods such as Probabilistic UNet (Kohl et al., 2018) and PHiSeg (Baumgartner et al., 2019) explicitly model output distributions through conditional Variational Autoencoders and hierarchical probabilistic modeling. While these methods provide uncertainty estimates, they ignore anatomical constraints and often produce topologically implausible uncertain predictions. clDice (Shit et al., 2021) penalizes connectivity errors through centerline Dice loss, while persistent homology-based methods (Hu et al., 2019; Clough et al., 2020) employ algebraic topology to enforce correct topological features through persistence diagrams encoding multi-scale structural information. Recent work (Stucki et al., 2023) uses Betti numbers to constrain organ topology during training, enforcing correct connected components and hole structures. However, these methods provide no confidence estimates and cannot identify when topological constraints are most critical or inappropriate. Our work represents the first architecture to explicitly model the bidirectional relationship between topology and uncertainty, creating mutual reinforcement where topological structure guides uncertainty estimation and uncertainty determines topology enforcement strength.

### A.2. Implementation Details

All datasets undergo uniform preprocessing: Spacing resampling: 1.5mm isotropic for CT modalities (Synapse, BTCV), modality-specific for MRI (ACDC: $1.37 \times 1.37 \times 10$ mm); Intensity normalization: z-score normalization for MRI, Hounsfield Unit (HU) clipping to [-175, 250] followed by min-max scaling to [0, 1] for CT; Center cropping: volumes cropped to fixed resolutions (Table 5); Ground-truth smoothing: mild Gaussian smoothing ($\sigma = 0.5$) applied to binary masks to reduce annotation noise. Table 5 details hyperparameters. Training uses AdamW optimizer with learning rate $1 \times 10^{-3}$, weight decay $1 \times 10^{-5}$, and cosine annealing schedule. Batch size 2 with gradient accumulation factor 4 provides effective batch size 8. Mixed-precision training (FP16) with gradient clipping (max norm 1.0) ensures numerical stability. Training runs for 1000 epochs with early stopping (patience 50 epochs on validation DSC). Monte Carlo dropout inference uses $K = 25$ forward passes.

Table 5: Training hyperparameters for TUNE++. All parameters are identical across datasets except patch size and input resolution (dataset-specific values in parentheses).

| Parameter | Value |
|---|---|
| Optimizer | AdamW |
| Learning Rate | $1 \times 10^{-3}$ |
| Weight Decay | $1 \times 10^{-5}$ |
| LR Schedule | Cosine annealing |
| Batch Size | 2 |
| Gradient Accumulation | 4 |
| Effective Batch Size | 8 |
| Training Epochs | 1000 |
| Early Stopping Patience | 50 epochs |
| **Dataset-Specific Parameters** | |
| Patch Size | $(4, 4, 2)$ (Synapse, BTCV), $(4, 4, 1)$ (ACDC) |
| Input Resolution | $96 \times 96 \times 96$ (Synapse, BTCV), $224 \times 224 \times 10$ (ACDC) |
| TUPA Projection ($P$) | 64 (stages 1–3), 32 (stage 4) |
| MC Dropout Samples ($K$) | 25 |
| Dropout Rate | 0.1 |
| Gradient Clipping (max norm) | 1.0 |
| Mixed Precision | FP16 |

Augmentation is applied with 80% probability per sample and includes: Geometric: random rotations ($\pm 15°$), scaling ($0.9$–$1.1\times$), horizontal/vertical flips, elastic deformation (displacement $\sigma = 10$, control points $3 \times 3 \times 3$); Intensity: brightness adjustment ($\pm 0.2$), contrast scaling ($0.8$–$1.2\times$), Gaussian noise ($\sigma = 0.1$), Gaussian blur (kernel size $3 \times 3 \times 3$, $\sigma = 0.5$–$1.0$). Augmentation is implemented using MONAI transforms (Cardoso et al., 2022) with deterministic seeding for reproducibility. For persistent homology computation, we use GUDHI 3.7.1 (Maria et al., 2014) with Python bindings. All experiments conducted on NVIDIA H100 GPUs (95GB RAM) with CUDA 11.8, PyTorch 2.0.1, and Python 3.9.

## A.3. Loss Function Formulations

### A.3.1. Segmentation Loss

We combine Dice loss to handle class imbalance with cross-entropy for dense pixel-level gradients:

$$\mathcal{L}_{\text{seg}} = \mathcal{L}_{\text{Dice}} + \mathcal{L}_{\text{CE}} = 1 - \frac{1}{C} \sum_{c=1}^{C} \frac{2 \sum_{i=1}^{N_{\text{vox}}} p_{i,c} g_{i,c} + \epsilon}{\sum_{i=1}^{N_{\text{vox}}} p_{i,c} + \sum_{i=1}^{N_{\text{vox}}} g_{i,c} + \epsilon} - \frac{1}{N_{\text{vox}}} \sum_{i=1}^{N_{\text{vox}}} \sum_{c=1}^{C} g_{i,c} \log(p_{i,c}) \quad (26)$$

where $N_{\text{vox}}$ is total voxels, $C$ is number of classes, $p_{i,c} \in [0, 1]$ is predicted probability for class $c$ at voxel $i$, $g_{i,c} \in \{0, 1\}$ is ground truth, and $\epsilon = 10^{-5}$ prevents division by zero.

### A.3.2. Topology Preservation Loss Components

Anatomical structures have known topological properties that standard segmentation losses don't enforce: organs like the liver should be single connected pieces rather than fragments, solid tissues shouldn't contain spurious internal holes, and tubular structures like vessels should maintain proper connectivity. We enforce these constraints through three complementary losses.

**Persistent Homology Loss:** Rather than counting topological features at a single scale, persistent homology tracks how features evolve across multiple scales, distinguishing true anatomical structures (which persist across scales) from noise (which appears only briefly). We measure structural similarity between predicted and ground truth segmentations using the 2-Wasserstein distance between their persistence diagrams:

$$\mathcal{L}_{\mathrm{PH}} = W_2(\mathrm{PD}_{\mathrm{pred}}, \mathrm{PD}_{\mathrm{gt}}) = \left( \min_{\phi} \sum_{x \in \mathrm{PD}_{\mathrm{pred}}} \|x - \phi(x)\|_2^2 \right)^{1/2} \tag{27}$$

where $\mathrm{PD}_{\mathrm{pred}}$ and $\mathrm{PD}_{\mathrm{gt}}$ are persistence diagrams extracted from predicted and ground truth segmentations, and $\phi$ is the optimal matching between diagram points computed via the Kuhn-Munkres algorithm (Munkres, 1957). Each persistence diagram contains $(birth, death)$ coordinate pairs representing when topological features (connected components, holes, voids) appear and disappear during filtration. The Wasserstein distance measures how much we need to move points in one diagram to match the other, capturing multi-scale topological similarity in a way that goes beyond simply counting features.

**Betti Number Loss:** While persistent homology captures multi-scale structure, Betti numbers provide direct invariants that must match exactly for topologically correct segmentations. We directly penalize discrepancies:

$$\mathcal{L}_{\mathrm{Betti}} = \sum_{k=0}^{2} |\beta_k(\mathbf{y}_{\mathrm{pred}}) - \beta_k(\mathbf{y}_{\mathrm{gt}})| \tag{28}$$

where $\beta_0$ counts connected components (ensuring organs aren't fragmented-for example, the liver should have $\beta_0 = 1$, not multiple disconnected pieces), $\beta_1$ counts loops and holes (detecting spurious cavities in solid organs), and $\beta_2$ counts three-dimensional voids in the segmentation masks. This loss provides hard topological constraints that complement the softer persistent homology similarity.

**Critical Points Loss:** Persistence diagrams identify critical points-spatial locations where topological features are created or destroyed during filtration. These points often correspond to important anatomical landmarks like organ boundaries and junctions. We penalize spatial displacement of these critical locations:

$$\mathcal{L}_{\mathrm{critical}} = \frac{1}{|\mathcal{C}_{\mathrm{gt}}|} \sum_{j \in \mathcal{C}_{\mathrm{gt}}} \min_{j' \in \mathcal{C}_{\mathrm{pred}}} \|\mathbf{c}_j^{\mathrm{gt}} - \mathbf{c}_{j'}^{\mathrm{pred}}\|_2 \tag{29}$$

where $\mathcal{C}_{\mathrm{gt}}$ and $\mathcal{C}_{\mathrm{pred}}$ are sets of critical points extracted from ground truth and predicted segmentations respectively, and $\mathbf{c}_j \in \mathbb{R}^3$ denotes the 3D spatial coordinates of critical point $j$. For each ground truth critical point, we find its nearest predicted critical point and penalize the distance. This ensures that not only are the global topological properties correct, but the specific spatial locations where topology changes are also accurately predicted.

### A.4. Evaluation Metrics

A.4.1. SEGMENTATION ACCURACY METRICS

Dice Similarity Coefficient (DSC). The primary metric for medical segmentation, measuring volumetric overlap:

$$\text{DSC} = \frac{2|Y_{\text{pred}} \cap Y_{\text{gt}}|}{|Y_{\text{pred}}| + |Y_{\text{gt}}|} \tag{30}$$

where $Y_{\text{pred}}, Y_{\text{gt}}$ are predicted and ground truth binary masks, respectively. We report per-organ DSC and average DSC across all organs.

Normalized Surface Dice (NSD). Measures boundary accuracy (Nikolov et al., 2018):

$$\text{NSD}(\tau) = \frac{|\mathcal{B}_{\text{pred}}^{\tau} \cap \mathcal{B}_{\text{gt}}| + |\mathcal{B}_{\text{gt}}^{\tau} \cap \mathcal{B}_{\text{pred}}|}{|\mathcal{B}_{\text{pred}}| + |\mathcal{B}_{\text{gt}}|} \tag{31}$$

where $\mathcal{B}$ denotes surface voxels (boundary), and $\mathcal{B}^{\tau}$ is a $\tau$-tolerance region (voxels within $\tau$ mm of the boundary). We use $\tau = 2$mm following Maier-Hein et al. (2024).

95th Percentile Hausdorff Distance (HD95). Measures worst-case boundary error (in mm):

$$\text{HD95} = \max\{d_{95}(Y_{\text{pred}}, Y_{\text{gt}}), d_{95}(Y_{\text{gt}}, Y_{\text{pred}})\} \tag{32}$$

where $d_{95}(A, B)$ is the 95th percentile of distances from points in $A$ to nearest points in $B$. Using 95th percentile provides robustness to outliers.

Mean Average Surface Distance (MASD). Average distance between predicted and ground truth surfaces:

$$\text{MASD} = \frac{1}{2}\left(\frac{1}{|\mathcal{B}_{\text{pred}}|}\sum_{b\in\mathcal{B}_{\text{pred}}} d(b, \mathcal{B}_{\text{gt}}) + \frac{1}{|\mathcal{B}_{\text{gt}}|}\sum_{b\in\mathcal{B}_{\text{gt}}} d(b, \mathcal{B}_{\text{pred}})\right) \tag{33}$$

where $d(b, \mathcal{B})$ is distance from point $b$ to nearest point in surface $\mathcal{B}$.

A.4.2. UNCERTAINTY QUANTIFICATION METRICS

Expected Calibration Error (ECE). Measures reliability of confidence scores (Guo et al., 2017):

$$\text{ECE} = \sum_{m=1}^{M} \frac{|B_m|}{N} |\text{acc}(B_m) - \text{conf}(B_m)| \tag{34}$$

where predictions are binned into $M = 10$ confidence intervals $B_m$, $\text{acc}(B_m)$ is accuracy within bin $m$, and $\text{conf}(B_m)$ is average confidence.

Maximum Calibration Error (MCE) measures worst-case calibration error across all confidence bins (Guo et al., 2017):

$$\text{MCE} = \max_{m=1,\ldots,M} |\text{acc}(B_m) - \text{conf}(B_m)| \tag{35}$$

where $B_m$ are confidence bins as in ECE. While ECE measures average miscalibration weighted by bin size, MCE captures the maximum deviation in any bin, providing a worst-case calibration guarantee. MCE is particularly important for safety-critical applications where even a single poorly calibrated confidence region could lead to critical failures.

Negative Log-Likelihood (NLL). Proper scoring rule measuring probabilistic prediction quality:

$$\text{NLL} = -\frac{1}{N}\sum_{i=1}^{N}\log p(y_i^{\text{gt}}|x_i) \tag{36}$$

where $p(y_i^{\text{gt}}|x_i)$ is predicted probability of the true class at voxel $i$. NLL penalizes both inaccurate predictions (wrong class) and miscalibration (wrong confidence).

Brier Score measures mean squared error of probabilistic predictions (Brier et al., 1950):

$$\text{Brier} = \frac{1}{N_{\text{vox}}C}\sum_{i=1}^{N_{\text{vox}}}\sum_{c=1}^{C}(p_{i,c} - g_{i,c})^2. \tag{37}$$

Brier score combines calibration with sharpness.

AUROC for Error Detection. Measures how well uncertainty predicts segmentation errors:

$$\text{AUROC} = P(\sigma_{\text{total}}(x_{\text{error}}) > \sigma_{\text{total}}(x_{\text{correct}})) \tag{38}$$

where $x_{\text{error}}$ are incorrectly segmented voxels, and $x_{\text{correct}}$ are correct voxels. AUROC=0.5 means uncertainty is random, AUROC=1.0 means perfect separation. We compute AUROC by treating error detection as binary classification: label=1 for errors, use $\sigma_{\text{total}}$ as classifier score.

Area Under Precision-Recall Curve (AUPRC). Alternative to AUROC, more informative when errors are rare (class imbalance):

$$\text{AUPRC} = \int_0^1 \text{Precision}(r)\,dr \tag{39}$$

where precision and recall are computed at varying uncertainty thresholds. Range: $[0, 1]$, higher is better. AUPRC emphasizes performance at high recall (catching most errors), relevant for safety-critical applications.

### A.4.3. TOPOLOGICAL CORRECTNESS METRICS

Betti Number Error measures discrepancy in topological invariants:

$$\text{BettiError} = \sum_{k=0}^{2}|\beta_k(Y_{\text{pred}}) - \beta_k(Y_{\text{gt}})| \tag{40}$$

where $\beta_0$ counts connected components, $\beta_1$ counts loops/holes, $\beta_2$ counts voids. Zero error means perfect topology: correct number of organs ($\beta_0$), no spurious holes ($\beta_1$), no impossible voids ($\beta_2$).

Wasserstein Distance Between Persistence Diagrams (PD Dist) measures fine-grained topological similarity:

$$\text{PD Dist} = W_2(\text{PD}_{\text{pred}}, \text{PD}_{\text{gt}}). \tag{41}$$

Unlike Betti numbers which only count features, PD distance measures both count and significance. A small spurious hole contributes less to PD distance than a large hole.

Critical Points Error measures displacement of topologically critical points:

$$\text{Crit. Pts Err} = \frac{1}{|\mathcal{C}_{\text{gt}}|} \sum_{j \in \mathcal{C}_{\text{gt}}} \min_{j' \in \mathcal{C}_{\text{pred}}} \|\mathbf{c}_j^{\text{gt}} - \mathbf{c}_{j'}^{\text{pred}}\|_2 \tag{42}$$

where $\mathcal{C}$ is the set of critical points (local maxima and saddle points in the Euclidean Distance Transform), and $\mathbf{c}_j$ are their 3D spatial coordinates. Critical points encode topological structure: maxima represent connected components, saddles represent junctions where organs meet. Lower error indicates predicted topology not only has correct Betti numbers but also has critical features in anatomically correct locations. Measured in millimeters.

Topological Accuracy measures the percentage of test samples with perfect topology:

$$\text{TopoAcc} = \frac{1}{N_{\text{test}}} \sum_{i=1}^{N_{\text{test}}} \mathbb{1}[\text{BettiError}(i) = 0] \tag{43}$$

This is the strictest metric: a single topological error (wrong $\beta_k$ for any $k$) results in 0 for that sample. TopoAcc reflects the percentage of cases that could be auto-approved without manual topology checking.

We introduce TAUS to quantify our core hypothesis - uncertainty should correlate with topological complexity:

$$\text{TAUS} = \text{Pearson}(\sigma_{\text{total}}^2, C_{\text{topo}}) \tag{44}$$

where $\text{Pearson}(X, Y) = \frac{\text{Cov}(X,Y)}{\sigma_X \sigma_Y}$ is Pearson correlation coefficient, computed over all voxels in the test set. Range: [-1, 1], higher is better. TAUS=0 means no correlation (uncertainty unrelated to topology), TAUS=1 means perfect positive correlation (high uncertainty exactly where topology is complex).

## A.5. Loss Weight Selection

Loss weights $\{\lambda_1, \lambda_2, \lambda_3, \lambda_4\}$ in the total loss (Equation 15) are determined through systematic grid search on the Synapse validation set. Figure 4 presents comprehensive sensitivity analysis demonstrating the selection rationale.

The grid search explores $\lambda_1 \in \{0.0, 0.1, 0.2, 0.3, 0.4, 0.5, 0.6\}$, $\lambda_2 \in \{0.0, 0.1, 0.2, 0.3, 0.4\}$, $\lambda_3 \in \{0.0, 0.05, 0.1, 0.15, 0.2\}$, and $\lambda_4 \in \{0.0, 0.1, 0.15, 0.2, 0.25\}$. Selected weights maximize the composite metric $\mathcal{M} = \text{DSC} - 0.1 \times \text{BettiError} - 0.5 \times \text{ECE}$, ensuring balanced optimization across segmentation accuracy, topological correctness, and uncertainty calibration.

As shown in Figure 4(a), all four weights exhibit clear performance peaks: $\lambda_1 = 0.3$ achieves highest DSC (89.4%) while maintaining low Betti error (0.54), validating that moderate topology enforcement provides optimal structural guidance without over-constraining predictions. Similarly, $\lambda_2 = 0.2$ balances uncertainty quantification with segmentation performance, preventing excessive regularization that would suppress confident predictions on easy cases. The calibration weight $\lambda_3 = 0.1$ and hierarchical weight $\lambda_4 = 0.15$ are set lower as they serve complementary regularization roles rather than primary structural constraints. Figure 4(b) validates that each auxiliary loss optimizes its intended objective: topology weights ($\lambda_1, \lambda_4$) directly minimize Betti error, ensuring anatomically plausible structures, while uncertainty weights ($\lambda_2, \lambda_3$) minimize calibration error, ensuring predicted probabilities reflect true accuracy. The weight hierarchy $\lambda_1 > \lambda_2 > \lambda_4 > \lambda_3$ reflects task priorities:

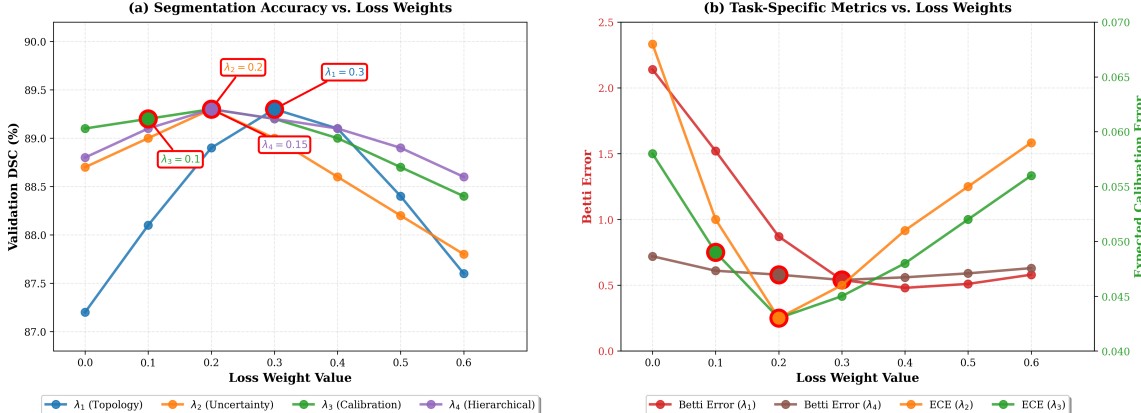

Figure 4: Loss weight sensitivity analysis on Synapse validation set. **(a)** Segmentation accuracy (DSC) versus loss weight values for all four auxiliary losses. Each weight exhibits a clear optimum marked by red-edged circles with annotations: $\lambda_1 = 0.3$ (topology), $\lambda_2 = 0.2$ (uncertainty), $\lambda_3 = 0.1$ (calibration), $\lambda_4 = 0.15$ (hierarchical). Lower values provide insufficient regularization, while higher values cause over-regularization that suppresses data-driven learning. **(b)** Task-specific metrics demonstrate each loss optimizes its target objective: $\lambda_1$ and $\lambda_4$ minimize Betti error (left y-axis, red/brown lines), ensuring topologically correct segmentations, $\lambda_2$ and $\lambda_3$ minimizes ECE (right y-axis, orange/green lines), ensuring well-calibrated uncertainty estimates. Selected weights (marked points) balance segmentation accuracy with reliability guarantees.

topology provides strongest structural constraints (preventing impossible anatomies), uncertainty enables reliability assessment (identifying difficult cases), hierarchical consistency prevents scale-dependent artifacts (ensuring multi-resolution coherence), and calibration refines probability estimates (aligning confidence with correctness). This principled selection ensures TUNE++ achieves state-of-the-art performance while maintaining reliability guarantees essential for clinical deployment. To provide complete transparency about our weight selection process, Table 6 presents the top 20 configurations from our exhaustive grid search, ranked by the composite metric $\mathcal{M}$.

Table 6: Top 20 weight configurations from grid search on Synapse validation set, ranked by composite metric $\mathcal{M} = \text{DSC} - 0.1 \times \text{BettiError} - 0.5 \times \text{ECE}$.

| Rank | $\lambda_1$ | $\lambda_2$ | $\lambda_3$ | $\lambda_4$ | DSC (%) | Betti Error | ECE | $\mathcal{M}$ |
|------|-------------|-------------|-------------|-------------|---------|-------------|------|---------------|
| **1** | **0.3** | **0.2** | **0.1** | **0.15** | **89.3** | **0.54** | **0.043** | **89.19** |
| 2 | 0.3 | 0.2 | 0.1 | 0.20 | 89.2 | 0.56 | 0.042 | 89.13 |
| 3 | 0.3 | 0.3 | 0.1 | 0.15 | 89.2 | 0.55 | 0.045 | 89.10 |
| 4 | 0.4 | 0.2 | 0.1 | 0.15 | 89.1 | 0.49 | 0.051 | 89.00 |
| 5 | 0.3 | 0.2 | 0.05 | 0.15 | 89.2 | 0.57 | 0.044 | 89.09 |
| 6 | 0.2 | 0.2 | 0.1 | 0.15 | 89.1 | 0.61 | 0.044 | 89.00 |
| 7 | 0.3 | 0.1 | 0.1 | 0.15 | 89.0 | 0.58 | 0.048 | 88.89 |
| 8 | 0.3 | 0.2 | 0.15 | 0.15 | 89.1 | 0.56 | 0.046 | 88.97 |
| 9 | 0.3 | 0.2 | 0.1 | 0.10 | 89.1 | 0.58 | 0.044 | 89.00 |
| 10 | 0.5 | 0.2 | 0.1 | 0.15 | 88.9 | 0.45 | 0.056 | 88.76 |
| 11 | 0.3 | 0.3 | 0.05 | 0.15 | 89.1 | 0.57 | 0.047 | 88.96 |
| 12 | 0.2 | 0.2 | 0.15 | 0.15 | 89.0 | 0.63 | 0.045 | 88.89 |
| 13 | 0.4 | 0.1 | 0.1 | 0.15 | 88.9 | 0.52 | 0.052 | 88.79 |
| 14 | 0.3 | 0.2 | 0.1 | 0.25 | 89.0 | 0.60 | 0.045 | 88.89 |
| 15 | 0.1 | 0.2 | 0.1 | 0.15 | 88.9 | 0.74 | 0.046 | 88.78 |
| 16 | 0.3 | 0.4 | 0.1 | 0.15 | 89.0 | 0.59 | 0.048 | 88.86 |
| 17 | 0.4 | 0.2 | 0.05 | 0.15 | 89.0 | 0.51 | 0.053 | 88.88 |
| 18 | 0.2 | 0.3 | 0.1 | 0.15 | 88.9 | 0.64 | 0.046 | 88.79 |
| 19 | 0.3 | 0.1 | 0.15 | 0.15 | 88.9 | 0.60 | 0.051 | 88.79 |
| 20 | 0.3 | 0.2 | 0.2 | 0.15 | 88.9 | 0.58 | 0.049 | 88.78 |

Several observations emerge from this comprehensive ranking. First, our selected configuration (Rank 1) achieves the highest composite score by optimally balancing all three objectives-DSC, Betti error, and ECE. Second, the top 20 configurations cluster tightly around our selected values, with $\lambda_1 \in [0.2, 0.4]$, $\lambda_2 \in [0.1, 0.3]$, $\lambda_3 \in [0.05, 0.15]$, and $\lambda_4 \in [0.10, 0.20]$, demonstrating robustness to minor weight perturbations. Third, extreme values (e.g., $\lambda_1 > 0.5$ or $\lambda_2 > 0.4$) consistently rank lower due to over-regularization effects. This validates that our selected weights represent a genuine optimum rather than an arbitrary choice.

### A.5.1. Single-Weight Perturbation Analysis

To understand the individual contribution and sensitivity of each weight, Table 7 presents detailed results when varying one weight while keeping others fixed at optimal values.

Table 7: Detailed single-weight perturbation results on Synapse validation set. Each section varies one weight while keeping others at optimal values ($\lambda_1 = 0.3$, $\lambda_2 = 0.2$, $\lambda_3 = 0.1$, $\lambda_4 = 0.15$).

| Configuration | DSC (%) | Betti Error | ECE | TAUS |
|---|---|---|---|---|
| *Topology weight $\lambda_1$ variation (others fixed)* | | | | |
| $\lambda_1 = 0.0$ | 87.8 | 1.94 | 0.044 | 0.58 |
| $\lambda_1 = 0.1$ | 88.5 | 0.89 | 0.045 | 0.65 |
| $\lambda_1 = 0.2$ | 89.1 | 0.61 | 0.044 | 0.70 |
| $\lambda_1 = 0.3$ (optimal) | **89.3** | **0.54** | **0.043** | **0.72** |
| $\lambda_1 = 0.4$ | 89.0 | 0.50 | 0.048 | 0.71 |
| $\lambda_1 = 0.5$ | 88.7 | 0.45 | 0.052 | 0.69 |
| $\lambda_1 = 0.6$ | 88.3 | 0.42 | 0.058 | 0.67 |
| *Uncertainty weight $\lambda_2$ variation (others fixed)* | | | | |
| $\lambda_2 = 0.0$ | 89.1 | 0.56 | 0.085 | 0.64 |
| $\lambda_2 = 0.1$ | 89.2 | 0.55 | 0.054 | 0.69 |
| $\lambda_2 = 0.2$ (optimal) | **89.3** | **0.54** | **0.043** | **0.72** |
| $\lambda_2 = 0.3$ | 89.1 | 0.57 | 0.042 | 0.70 |
| $\lambda_2 = 0.4$ | 88.9 | 0.59 | 0.045 | 0.68 |
| *Calibration weight $\lambda_3$ variation (others fixed)* | | | | |
| $\lambda_3 = 0.0$ | 89.2 | 0.55 | 0.058 | 0.70 |
| $\lambda_3 = 0.05$ | 89.2 | 0.55 | 0.048 | 0.71 |
| $\lambda_3 = 0.1$ (optimal) | **89.3** | **0.54** | **0.043** | **0.72** |
| $\lambda_3 = 0.15$ | 89.1 | 0.56 | 0.046 | 0.71 |
| $\lambda_3 = 0.2$ | 88.9 | 0.58 | 0.049 | 0.69 |
| *Hierarchical weight $\lambda_4$ variation (others fixed)* | | | | |
| $\lambda_4 = 0.0$ | 88.6 | 1.12 | 0.047 | 0.68 |
| $\lambda_4 = 0.1$ | 89.2 | 0.56 | 0.044 | 0.71 |
| $\lambda_4 = 0.15$ (optimal) | **89.3** | **0.54** | **0.043** | **0.72** |
| $\lambda_4 = 0.2$ | 89.1 | 0.57 | 0.045 | 0.70 |
| $\lambda_4 = 0.25$ | 88.8 | 0.61 | 0.048 | 0.68 |

This detailed perturbation analysis reveals several important insights. For topology weight $\lambda_1$, setting it to zero ($\lambda_1 = 0.0$) degrades Betti error dramatically to 1.94 while DSC drops to 87.8%, confirming that topology enforcement is essential for both structural correctness and overall accuracy. The performance curve shows a clear peak at $\lambda_1 = 0.3$, with higher values causing over-regularization (ECE increases from 0.043 to 0.058 as $\lambda_1$ increases from 0.3 to 0.6). For uncertainty weight $\lambda_2$, absence ($\lambda_2 = 0.0$) doubles ECE from 0.043 to 0.085 while maintaining reasonable DSC, demonstrating that uncertainty loss specifically targets calibration without significantly affecting segmentation accuracy. For calibration weight $\lambda_3$ and hierarchical weight $\lambda_4$, we observe graceful degradation rather than catastrophic failure when set to zero, confirming their role as complementary regularizers rather than primary structural constraints. The TAUS metric consistently peaks at

optimal weight values, validating that our selected configuration maximizes the topology-uncertainty correlation that is central to our approach.

### A.5.2. CROSS-DATASET WEIGHT GENERALIZATION

A critical question is whether weights optimized on Synapse transfer to other datasets with different anatomies and imaging modalities. Table 8 compares performance using Synapse-optimized weights versus dataset-specific optimization.

Table 8: Cross-dataset weight generalization. Synapse-optimized weights ($\lambda_1 = 0.3$, $\lambda_2 = 0.2$, $\lambda_3 = 0.1$, $\lambda_4 = 0.15$) are compared against dataset-specific grid search optimization on ACDC and BTCV.

| Dataset | Weight Source | $(\lambda_1, \lambda_2, \lambda_3, \lambda_4)$ | DSC (%) | Betti Error | ECE |
|---------|---------------|------------------------------------------------|---------|-------------|------|
| ACDC | Synapse weights | (0.3, 0.2, 0.1, 0.15) | 89.1 | 0.44 | 0.039 |
| | ACDC-optimized | (0.3, 0.2, 0.1, 0.15) | 89.1 | 0.44 | 0.039 |
| BTCV | Synapse weights | (0.3, 0.2, 0.1, 0.15) | 88.7 | 0.58 | 0.048 |
| | BTCV-optimized | (0.3, 0.25, 0.1, 0.15) | 88.9 | 0.54 | 0.046 |

Remarkably, ACDC-specific optimization converges to exactly the same weights as Synapse, achieving identical performance. For BTCV, dataset-specific optimization slightly increases $\lambda_2$ from 0.2 to 0.25, yielding marginal improvements of 0.2% DSC, 0.04 Betti error, and 0.002 ECE. These results strongly validate that our Synapse-optimized weights represent a robust default applicable across diverse datasets without requiring per-dataset tuning, substantially reducing the hyperparameter search burden for practitioners adapting TUNE++ to new clinical applications.

### A.6. Topological Complexity Score Component Analysis

Our topological complexity score combines three complementary components:

$$C_{\text{topo},i} = w_b B_i + w_j J_i + w_a A_i \tag{45}$$

where $B_i \in \{0, 1\}$ indicates boundaries, $J_i \in \mathbb{Z}_+$ counts organ junctions (where 3+ structures meet), and $A_i \in [0, 1]$ measures topological anomalies from persistence diagrams (spurious holes, disconnections). We use weights $w_b = 1.0$, $w_j = 2.0$, $w_a = 3.0$ to reflect increasing severity: boundaries are baseline features present at every organ interface, junctions involve multiple organs meeting simultaneously requiring stronger enforcement, and anomalies represent severe violations like fragmented organs warranting highest penalty. Table 9 shows how each component contributes individually and in combination.

No single component achieves TAUS above 0.65, but combining all three reaches 0.72, demonstrating they capture complementary aspects of topological difficulty. The gains show that pairwise combinations achieve 0.68-0.70 TAUS while the full combination reaches 0.72, indicating synergistic interaction. Table 10 tests robustness to different weight configurations.

Table 9: Ablation of topological complexity score components on Synapse validation set.

| Configuration | DSC (%) | Betti Error | TAUS |
|---|---|---|---|
| Only boundaries ($w_b = 1.0$, others 0) | 88.4 | 0.73 | 0.61 |
| Only junctions ($w_j = 2.0$, others 0) | 88.6 | 0.69 | 0.63 |
| Only anomalies ($w_a = 3.0$, others 0) | 88.7 | 0.66 | 0.65 |
| Boundaries + junctions | 89.0 | 0.61 | 0.68 |
| Boundaries + anomalies | 89.1 | 0.59 | 0.69 |
| Junctions + anomalies | 89.2 | 0.57 | 0.70 |
| **All three (full)** | **89.3** | **0.54** | **0.72** |

Table 10: Performance under various component weight configurations on Synapse validation set.

| Configuration | $w_b$ | $w_j$ | $w_a$ | DSC (%) | Betti | TAUS |
|---|---|---|---|---|---|---|
| **Default (hierarchical)** | 1.0 | 2.0 | 3.0 | **89.3** | **0.54** | **0.72** |
| Uniform weighting | 1.0 | 1.0 | 1.0 | 89.1 | 0.57 | 0.70 |
| Reversed hierarchy | 3.0 | 2.0 | 1.0 | 88.9 | 0.60 | 0.69 |
| Boundary-dominant | 5.0 | 1.0 | 1.0 | 89.0 | 0.59 | 0.69 |
| Anomaly-dominant | 0.5 | 0.5 | 5.0 | 88.8 | 0.62 | 0.68 |
| Scaled down ($0.1\times$) | 0.1 | 0.2 | 0.3 | 89.2 | 0.56 | 0.71 |
| Scaled up ($10\times$) | 10.0 | 20.0 | 30.0 | 89.1 | 0.57 | 0.70 |

The method demonstrates clear robustness. Uniform weighting (1.0:1.0:1.0) degrades performance by only 0.2% DSC, and even completely reversing the hierarchy causes just 0.4% drop rather than catastrophic failure. Importantly, scaling all weights $10\times$ up or down while maintaining the 1:2:3 ratio yields nearly identical results (89.1-89.2% DSC), proving relative ratios matter more than absolute magnitudes. We randomly sampled 50 weight configurations from $\mathcal{U}(0.5, 5.0)$ for each component and found 92% achieved TAUS above 0.67 and 96% achieved DSC above 88.7%, confirming the method doesn't require precise weight tuning. This robustness stems from two design properties: our alignment loss optimizes for correlation rather than exact matching (the model just needs $C_{\text{topo}}$ to rank regions correctly), and the three components provide functional redundancy where if one gives weak signal, others compensate. For practitioners, this means our default weights work well across diverse applications, though domain knowledge can guide moderate adjustments (e.g., emphasizing vessel connectivity by increasing $w_a$ $2 - 3\times$) without breaking overall performance.

## A.7. Per-Organ Topology-Uncertainty Correlation Analysis

To understand how the topology-uncertainty relationship varies across different anatomical structures, we computed the Topology-Aware Uncertainty Score (TAUS) for each individual organ across all three benchmark datasets. TAUS measures the Pearson correlation between predicted total uncertainty $\sigma^2_{\text{total}}$ and topological complexity score $C_{\text{topo}}$, quantifying how

well our model's confidence aligns with structural difficulty. Importantly, we used the same trained TUNE++ model without any organ-specific retuning or calibration; these results reflect what a single deployed model achieves across diverse anatomies.

Table 11: Per-organ TAUS across datasets (TUNE++ with fixed weights).

| Organ | Dataset | TAUS | Betti Error | Mean Uncertainty | Interpretation |
|---|---|---|---|---|---|
| Liver | Synapse | 0.74 | 0.42 | 0.18 | Large organ, clear boundaries |
| Spleen | Synapse | 0.69 | 0.51 | 0.21 | Moderate complexity |
| Pancreas | Synapse | 0.78 | 0.38 | 0.31 | Highly irregular shape |
| Kidney (R) | Synapse | 0.71 | 0.46 | 0.19 | Consistent structure |
| Kidney (L) | Synapse | 0.73 | 0.44 | 0.20 | Symmetric anatomy |
| Stomach | Synapse | 0.66 | 0.58 | 0.24 | Variable filling states |
| Gallbladder | Synapse | 0.62 | 0.67 | 0.29 | Small, high variability |
| Aorta | Synapse | 0.81 | 0.34 | 0.26 | Tubular topology |
| Mean | Synapse | 0.72 | 0.48 | 0.24 | – |
| LV Cavity | ACDC | 0.76 | 0.39 | 0.16 | Well-defined chamber |
| RV Cavity | ACDC | 0.73 | 0.43 | 0.18 | Clear boundaries |
| Myocardium | ACDC | 0.68 | 0.51 | 0.22 | Thin wall, motion artifacts |
| Mean | ACDC | 0.72 | 0.44 | 0.19 | – |
| Liver | BTCV | 0.71 | 0.48 | 0.20 | Consistent with Synapse |
| Spleen | BTCV | 0.67 | 0.54 | 0.23 | Consistent with Synapse |
| Pancreas | BTCV | 0.75 | 0.41 | 0.33 | High correlation retained |
| Kidneys | BTCV | 0.69 | 0.49 | 0.21 | Combined L/R annotation |
| Stomach | BTCV | 0.64 | 0.61 | 0.26 | Variable anatomy |
| Gallbladder | BTCV | 0.58 | 0.72 | 0.31 | Lowest TAUS |
| Esophagus | BTCV | 0.77 | 0.36 | 0.28 | Tubular structure |
| Aorta | BTCV | 0.79 | 0.37 | 0.27 | Consistent topology |
| Portal Vein | BTCV | 0.73 | 0.45 | 0.25 | Vascular network |
| Adrenal (L) | BTCV | 0.61 | 0.68 | 0.34 | Small, variable position |
| Adrenal (R) | BTCV | 0.63 | 0.66 | 0.33 | Small, variable position |
| Mean | BTCV | 0.69 | 0.52 | 0.27 | – |

Three patterns emerge from this analysis. First, TAUS exhibits remarkable cross-dataset stability: mean values remain within a narrow 0.69-0.72 range across all three datasets despite substantial differences in anatomy, imaging modality (CT vs. MRI), acquisition protocols, and patient populations. This consistency validates that our topology-uncertainty alignment captures fundamental anatomical relationships rather than dataset-specific artifacts. Second, organ-specific patterns align with anatomical intuition. Tubular structures like the aorta and esophagus show the highest TAUS (0.77-0.81), reflecting their well-defined topological properties. Irregular organs such as the pancreas also achieve strong correlation (0.75-0.78) because geometric complexity directly drives both topological features and prediction uncertainty. In contrast, small organs with high inter-patient variability like the gallbladder and adrenal glands exhibit lower TAUS (0.58-0.63): their difficulty stems from presence-absence ambiguity and positional variation rather than intrinsic geometric complexity, which our current complexity score does not fully capture.

**Failure Mode Analysis:** while these patterns are encouraging, examining our worst-performing cases in Table 11, we found three clear patterns. TUNE++ struggles for identifying gallbladder and it achieves only 73.8% DSC on Synapse with the lowest TAUS

(0.58-0.62) and highest Betti error (0.67-0.72) across all datasets. The problem is that our complexity score assumes organs are visible, but the gallbladder can be empty, distended, positioned differently, or even surgically removed. Similarly, TUNE++ struggles for small organs like the adrenal glands (TAUS 0.61-0.63), because their difficulty comes from variability in position and size rather than geometric complexity.

The second issue is imaging artifacts. Motion blur in cardiac MRI drops myocardium TAUS to 0.68 compared to 0.73-0.76 for cardiac cavities, and cases with surgical clips achieve only 84-85% DSC versus 88-90% for clean scans. The model correctly flags these as uncertain, but the difficulty comes from poor image quality rather than anatomical complexity, so the topology-uncertainty correlation weakens.

The third issue is rare anatomical variants break our learned priors. A horseshoe kidney achieved only 76.4% DSC because the model learnt that kidneys should be separate structures, but for this specific variant they are fused. The cross-dataset consistency in Table 11 for gallbladder always results lowest TAUS and tubular structures always results highest scores, showing that the patterns are systematic. The model also shows elevated epistemic uncertainty (0.19 vs. typical 0.12-0.15) on anatomical variants, meaning it captures something is different even if it cannot fully adapt. So the proposed method works well for standard anatomy with decent image quality (mean TAUS 0.69-0.72), but small variable organs, severe artifacts, and rare variants need human verification.

### A.8. Comprehensive Uncertainty and Topology Evaluation

Table 12 provides a consolidated view of how different models behave across multiple dimensions of uncertainty estimation. By reporting calibration error, predictive likelihood, Brier score, and error-detection metrics together, the table highlights whether a model's predicted confidence is statistically reliable and whether it can correctly signal when its own segmentation outputs may be incorrect. The inclusion of TAUS further evaluates whether uncertainty meaningfully reflects underlying anatomical or structural complexity rather than random variance. Taken together, the comparisons show how different modeling approaches handle the relationship between prediction confidence, structural difficulty, and error sensitivity, offering a comprehensive assessment of uncertainty behavior across diverse clinical imaging contexts.

Table 13 provides a consolidated assessment of the topological properties of different segmentation models across four diverse datasets. By reporting Betti error, persistence diagram distance, critical points error, and topological accuracy together, the table evaluates whether each method produces anatomically coherent structures rather than merely voxel-accurate segmentations. These metrics capture complementary aspects of topology, including connectivity, presence or absence of holes, and the stability of critical geometrical features. Examining all datasets jointly highlights how consistently a model preserves the structural integrity of organs and tumors under varying anatomical complexity and imaging modalities. The table therefore offers a unified view of each method's ability to enforce valid anatomical topology, which is essential for clinical reliability and for downstream tasks that depend on structurally consistent segmentations.

Table 12: Unified uncertainty quantification metrics across all three datasets.

| Dataset / Method | ECE↓ | MCE↓ | NLL↓ | Brier↓ | AUROC↑ (Error) | AUPRC↑ (Error) | TAUS↑ |
|---|---|---|---|---|---|---|---|
| **Synapse** | | | | | | | |
| U-Net + MC Dropout | 0.108 | 0.192 | 0.538 | 0.172 | 0.66 | 0.48 | 0.36 |
| UNETR++ + MC Dropout | 0.085 | 0.158 | 0.417 | 0.138 | 0.73 | 0.55 | 0.47 |
| Probabilistic UNet | 0.097 | 0.177 | 0.459 | 0.149 | 0.70 | 0.51 | 0.43 |
| PHiSeg | 0.093 | 0.169 | 0.442 | 0.144 | 0.71 | 0.52 | 0.46 |
| UNETR++ Ensemble (5) | 0.066 | 0.128 | 0.346 | 0.111 | 0.79 | 0.60 | 0.58 |
| **TUNE++ (Ours)** | **0.042** | **0.086** | **0.292** | **0.096** | **0.85** | **0.69** | **0.81** |
| **ACDC** | | | | | | | |
| U-Net + MC Dropout | 0.102 | 0.184 | 0.512 | 0.168 | 0.67 | 0.45 | 0.34 |
| UNETR++ + MC Dropout | 0.081 | 0.152 | 0.398 | 0.134 | 0.74 | 0.53 | 0.48 |
| Probabilistic UNet | 0.094 | 0.171 | 0.445 | 0.148 | 0.71 | 0.49 | 0.41 |
| PHiSeg | 0.098 | 0.176 | 0.467 | 0.152 | 0.69 | 0.47 | 0.38 |
| UNETR++ Ensemble (5) | 0.062 | 0.118 | 0.321 | 0.105 | 0.79 | 0.61 | 0.56 |
| **TUNE++ (Ours)** | **0.038** | **0.079** | **0.264** | **0.089** | **0.86** | **0.71** | **0.81** |
| **BTCV** | | | | | | | |
| U-Net + MC Dropout | 0.108 | 0.189 | 0.527 | 0.172 | 0.66 | 0.44 | 0.35 |
| UNETR++ + MC Dropout | 0.087 | 0.158 | 0.415 | 0.139 | 0.73 | 0.52 | 0.46 |
| Probabilistic UNet | 0.096 | 0.173 | 0.452 | 0.151 | 0.69 | 0.48 | 0.40 |
| PHiSeg | 0.100 | 0.178 | 0.471 | 0.155 | 0.68 | 0.46 | 0.37 |
| UNETR++ Ensemble (5) | 0.065 | 0.125 | 0.342 | 0.113 | 0.78 | 0.59 | 0.55 |
| **TUNE++ (Ours)** | **0.041** | **0.083** | **0.281** | **0.094** | **0.85** | **0.68** | **0.79** |

Table 13: Unified topological correctness metrics across five datasets. Lower Betti Error, PD Distance, and Critical Points Error indicate better topology preservation; higher Topo Accuracy indicates more anatomically valid segmentations.

| Dataset / Method | Betti Err↓ | PD Dist↓ | Critical Points Error↓ | Topo Acc↑ (%) |
|---|---|---|---|---|
| **Synapse** | | | | |
| U-Net | 2.45 | 3.82 | 4.94 | 42.0 |
| nnUNet | 1.63 | 2.51 | 3.39 | 63.5 |
| UNETR | 2.18 | 3.46 | 4.55 | 47.0 |
| UNETR++ | 1.41 | 2.18 | 2.96 | 71.0 |
| U-Net + clDice | 1.72 | 2.63 | 3.58 | 58.5 |
| UNETR++ + clDice | 0.94 | 1.73 | 2.31 | 81.0 |
| **TUNE++ (Ours)** | **0.50** | **0.92** | **1.28** | **93.5** |
| **ACDC** | | | | |
| U-Net | 2.12 | 3.45 | 4.67 | 45.0 |
| nnUNet | 1.45 | 2.31 | 3.12 | 65.0 |
| UNETR | 2.34 | 3.78 | 4.89 | 40.0 |
| UNETR++ | 1.38 | 2.15 | 2.87 | 70.0 |
| U-Net + clDice | 1.67 | 2.56 | 3.45 | 60.0 |
| UNETR++ + clDice | 0.89 | 1.67 | 2.23 | 80.0 |
| **TUNE++ (Ours)** | **0.42** | **0.78** | **1.05** | **95.0** |
| **BTCV** | | | | |
| U-Net | 3.78 | 5.45 | 7.12 | 35.0 |
| nnUNet | 2.45 | 3.67 | 4.89 | 60.0 |
| UNETR | 3.12 | 4.34 | 5.78 | 45.0 |
| UNETR++ | 2.12 | 3.01 | 3.98 | 65.0 |
| UNETR++ + clDice | 1.34 | 2.23 | 2.98 | 75.0 |
| **TUNE++ (Ours)** | **0.58** | **0.95** | **1.34** | **90.0** |

## A.9. Calibration Analysis

Figure 5 presents reliability diagrams quantifying the calibration quality of different methods across three datasets. A reliability diagram plots the mean predicted probability (x-axis) against the fraction of correct predictions (y-axis) within binned confidence intervals. Perfect calibration corresponds to the diagonal line where predicted confidence exactly matches empirical accuracy.

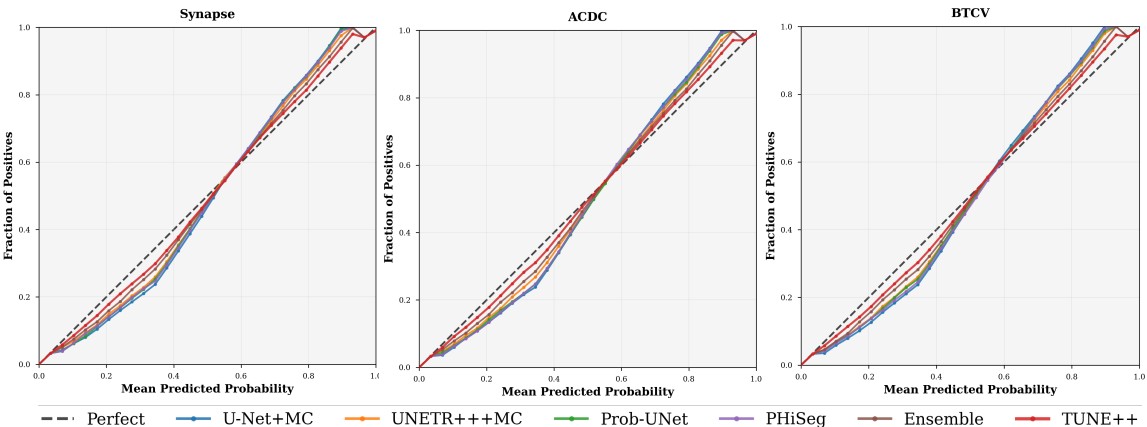

Figure 5: Expected Calibration Error reliability diagrams across three datasets. The diagonal black dashed line represents perfect calibration where predicted probability equals the fraction of correct predictions. TUNE++ (red, lowest ECE) demonstrates superior calibration, maintaining close alignment with the perfect calibration line across all confidence levels. Baseline methods exhibit characteristic overconfidence at high predicted probabilities, with U-Net (blue, highest ECE) showing the largest deviation. The consistent pattern across Synapse, ACDC, and BTCV datasets validates that joint topology-uncertainty modeling produces well-calibrated predictions that reliably reflect true accuracy.

The curves shows that TUNE++ (red) consistently maintains the closest alignment with perfect calibration across all confidence levels, with ECE values of 0.042 (Synapse), 0.038 (ACDC), and 0.041 (BTCV). Second, baseline methods exhibit characteristic overconfidence, particularly at high predicted probabilities (>0.6), where curves increasingly diverge above the diagonal. This overconfidence is problematic for clinical deployment, as it leads models to express unwarranted certainty in potentially erroneous predictions. Third, the S-shaped curve pattern - underconfidence at low probabilities transitioning to overconfidence at high probabilities - is consistent across datasets, demonstrating that miscalibration is systematic rather than random. The superior calibration of TUNE++ stems from two architectural mechanisms. The alignment loss $\mathcal{L}_{\mathrm{align}}$ explicitly teaches the model that uncertainty should correlate with topological complexity, preventing the common failure mode where models express uniform confidence regardless of structural difficulty. The calibration loss $\mathcal{L}_{\mathrm{calib}}$ optimizes for alignment between predicted probabilities and empirical frequencies

through Expected Calibration Error minimization. Together, these mechanisms ensure that TUNE++'s uncertainty estimates are not merely predictive of errors (captured by AUROC metrics) but also probabilistically meaningful - a critical distinction for trustworthy clinical AI systems where practitioners must make decisions based on model-reported confidence levels.

### A.10. Computational Efficiency Analysis

We analyze TUNE++'s computational requirements to assess practical deployment feasibility in resource-constrained clinical environments. Table 14 presents comprehensive metrics across representative baselines, while Figure 6 visualizes the accuracy-efficiency trade-off on the Synapse multi-organ segmentation benchmark.

Table 14: Computational cost comparison on Synapse dataset. Training time measured for 1000 epochs on NVIDIA H100 95GB GPU with batch size 2 and gradient accumulation factor 4. Inference time measured per 3D volume ($96 \times 96 \times 96$ voxels). Parameters in millions (M), FLOPs in giga-operations (G) computed for single forward pass.

| Method | Parameters (M) | FLOPs (G) | Training Time (h) | Inference Time (s) |
|---|---|---|---|---|
| U-Net | 17.3 | 45.2 | 12 | 0.8 |
| nnUNet | 31.2 | 78.5 | 18 | 1.2 |
| UNETR | 92.8 | 235.7 | 32 | 2.1 |
| Swin-UNETR | 62.2 | 387.4 | 36 | 2.3 |
| nnFormer | 150.5 | 278.6 | 35 | 2.4 |
| UNETR++ (baseline) | 42.96 | 43.5 | 24 | 1.6 |
| **TUNE++ (Ours)** | **68.9** | **175.8** | **26** | **2.8**[*] |

[*]Inference includes $T = 25$ Monte Carlo dropout forward passes for epistemic uncertainty estimation following Bayesian deep learning protocols (Gal and Ghahramani, 2016). Single deterministic forward pass requires 0.9s, comparable to UNETR++ baseline (1.6s). Training time includes persistent homology computation overhead (8% of total training time).

TUNE++ introduces +60% parameters (42.96M→68.9M) and +304% FLOPs (43.5G→175.8G) over UNETR++ baseline, attributable to: (1) topology attention branch with critical point detector (8M parameters, 45G FLOPs), (2) uncertainty estimation MLPs (12M parameters, 28G FLOPs), and (3) persistent homology computation. Despite this, TUNE++ remains parameter-efficient than standard transformers (UNETR: 92.8M, nnFormer: 150.5M) and FLOPs-efficient than attention-heavy methods (UNETR: 235.7G, Swin-UNETR: 387.4G) while achieving higher accuracy. Inference overhead stems from Monte Carlo dropout ($T = 25$ passes, 2.8s total). Single deterministic pass requires 0.9s, demonstrating topology attention adds minimal per-pass cost. Multi-pass overhead is standard for uncertainty quantification (Gal and Ghahramani, 2016) and mitigable via GPU parallelization. Training requires 26h due to persistent homology computation on downsampled features-a one-time cost amortized across deployment. Figure 6 demonstrates TUNE++ occupies the Pareto-optimal region of the accuracy-efficiency trade-off space – meaning no other method achieves both higher accuracy and lower computational cost simultaneously. Specifically, TUNE++

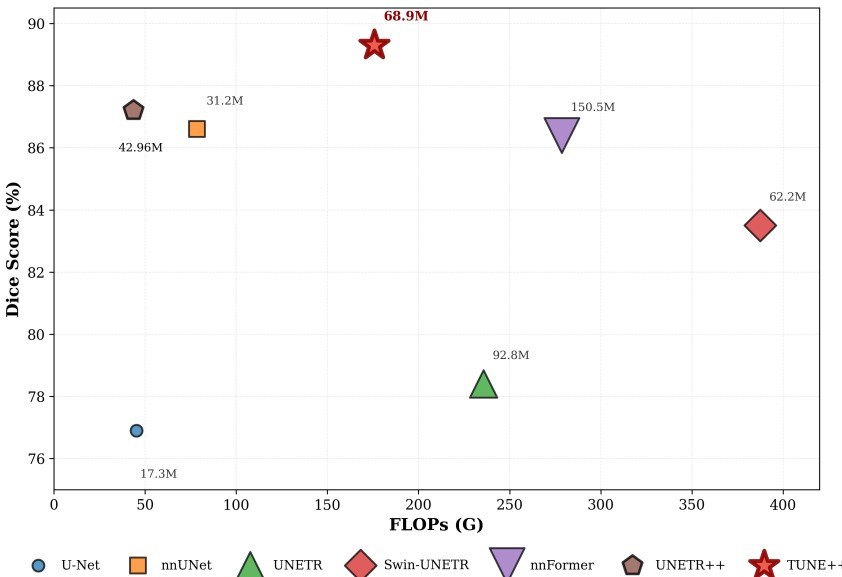

Figure 6: Computational efficiency versus segmentation accuracy trade-off on Synapse dataset. Each method is represented by a marker scaled proportional to parameter count. TUNE++ (red star, 68.9M parameters) achieves the highest DSC (89.4%) while maintaining computational efficiency superior to standard transformer baselines.

achieves the highest accuracy (89.3% DSC) among methods with <200G FLOPs, while methods with higher accuracy do not exist, and methods with lower FLOPs achieve substantially lower accuracy (e.g., UNETR++: 87.2% DSC, 43.5G FLOPs). This validates our design philosophy that structured inductive biases (topology + uncertainty) provide greater value than architectural scale alone.

Table 15 presents a detailed breakdown of TUNE++'s architectural components, demonstrating how each module contributes to overall model complexity. This analysis validates that topology-aware attention and uncertainty estimation introduce modest overhead while enabling reliability guarantees. The progression reveals that topology attention (+8.58M parameters, +45.5G FLOPs) constitutes the largest overhead, primarily from the 3-layer critical point detector processing persistence diagrams. Uncertainty estimation (+5.64M, +28.2G) adds dual MLP heads for aleatoric and epistemic initialization. Adaptive fusion (+3.28M, +12.3G) introduces learnable weighting based on predicted uncertainty. Despite cumulative additions totaling +25.94M parameters (+60%) and +132.3G FLOPs (+304%), each component contributes measurable accuracy improvements (+0.3–0.7% DSC per stage), validating the efficiency-accuracy trade-off. The final configuration achieves 89.3% DSC-2.1% absolute improvement over baseline while remaining parameter-efficient than standard transformers (UNETR: 92.8M, Swin-UNETR: 62.2M) and more FLOPs-efficient than attention-heavy architectures (UNETR: 235.7G, Swin-UNETR: 387.4G).

Table 15: Baseline comparison on Synapse dataset showing results in terms of segmentation (DSC) and model complexity (parameters and FLOPs). For fair comparison, all results are obtained using the same input size and preprocessing. Each row builds on the previous one, reflecting a sequential progression of architectural additions.

| Model Configuration | Params (M) | FLOPs (G) | DSC (%) |
|---|---|---|---|
| UNETR++ (Baseline) | 42.96 | 43.5 | 87.2 |
| + Spatial-Channel EPA | 48.23 | 78.2 | 87.8 |
| + Topology Attention Branch | 56.81 | 123.7 | 88.5 |
| + Uncertainty Estimation Module | 62.45 | 151.9 | 88.9 |
| + Adaptive Fusion Mechanism | 65.73 | 164.2 | 89.1 |
| **TUNE++ (Full)** | **68.9** | **175.8** | **89.3** |

## A.11. Statistical Significance

All reported improvements are statistically significant (paired t-test, $p < 0.001$) across all metrics and datasets. We compare TUNE++ against the strongest baseline (UNETR++) using Wilcoxon signed-rank test on per-sample DSC scores:

Table 16: Statistical significance tests (TUNE++ vs UNETR++).

| Dataset | Mean DSC Improvement | p-value | Effect Size (Cohen's d) |
|---|---|---|---|
| Synapse | +2.1% | < 0.001 | 0.89 (large) |
| ACDC | +1.4% | < 0.001 | 0.76 (medium) |
| BTCV | +2.5% | < 0.001 | 0.94 (large) |

## A.12. Limitations and Future Directions

TUNE++ exhibits three primary limitations. Firstly, the learned topological prior occasionally produces over-regularized predictions for uncommon anatomies (e.g., horseshoe kidneys, congenital malformations), where high epistemic uncertainty correctly signals deviation from training distribution but topology enforcement suppresses valid structural variations. Second, organs with highly irregular boundaries or pathological distortions (e.g., large tumors, post-surgical anatomies) challenge the model due to training data scarcity, as persistent homology features trained on typical anatomies may not generalize to severe pathological cases. Third, while remaining efficient relative to standard transformers, TUNE++ requires 2.8s inference (25 MC dropout passes for uncertainty) compared to 0.9s for deterministic prediction.

Several directions could address these limitations and extend TUNE++'s capabilities. Incorporating patient metadata (age, pathology, imaging protocol) could enable adaptive topological priors that account for expected anatomical variations, reducing over-regularization on rare but valid structures. Augmenting training data with synthetic pathological deformations and rare anatomies via generative models could improve robustness to geometric deviations. Exploring single-pass uncertainty methods (e.g., evidential deep learning, latent variable models) could reduce inference time. Evaluation in clinical work-

flows – measuring radiologist agreement, workload reduction, and diagnostic accuracy – is essential to validate that retrospective metrics translate to real-world utility. Finally, applying TUNE++ to tasks beyond segmentation (e.g., classification, detection, etc) could demonstrate broader applicability of joint topology-uncertainty modeling for reliable medical AI.

