# OpenReview forum: "TUNE++: Topology-Guided Uncertainty Estimation for Reliable 3D Medical Image Segmentation"
_MIDL.io/2026/Conference — MIDL 2026 Poster_

### Official Review · Reviewer_hAdr · 2026-01-08

**Confidence:** 2
**Preliminary Rating:** 3
**Final Rating:** 4

**Summary:**

The paper introduces a unified deep learning framework designed to improve the reliability of medical image segmentations by jointly modeling anatomical correctness (topology) and prediction confidence (uncertainty).
The authors compared segmentation performance on three datasets (Synapse, ACDC, and BTCV) and showed that their approach yields highest dice score and lowest calibration error.

**Strengths:**

The paper has a thorough evaluation, fair baseline models and a good number of datasets where they evaluated their method on.
Their results are convincing and their method section seems to be very thorough.

**Weaknesses:**

My main problem with the paper is that it is very hard to read, especially the methods section. You don't always properly define variables at the times used, for example in equation 1, the loss terms are not properly introduced when introducing the whole formula. That makes me jump back and forth so that I understand the method. In 2.1 for example, you use the variable C_1=32 without mentioning what the subscript 1 means. Moreover, in 2.1 you start talking about persistence diagrams without mentioning why they are useful or necessary for your method. I could keep going on like this for the methods section. It lacks a lot of structure. Before you do any derivations or introduce variables, you should help the reader understanding your reasoning instead of just writing context-less formulas.

**Detailed Comments:**

Please see points above

**Justification Of Final Rating:**

The authors have put in a significant amount of work to make the paper more readable. Especially the Methods section is now way easier to understand and appears to be more structured. My main critique points have been addressed and therefore I increased my score from borderline to "Weak accept".

**Justification Of The Preliminary Rating:**

I am willing to increase my score upon re-structuring the methods section. I like the core idea and the results are convincing but it is just really hard to read. I am not sure whether I still fully understand the paper without spending several more hours on it but for a conference paper I think this shouldn't be the case.

**Questions To Address In The Rebuttal:**

In order to make me improve my score, you need to adapt the methods section significantly. I would say that I'm not the dumbest person but I really had a hard time understanding your methods section because many parts are not properly defined or explained. Detailed derivations belong in a conference paper into the appendix. As there is anyways limited space you should focus on conveying the ideas and concepts.

---

> ### Author Response · Authors · 2026-01-25
>
> We sincerely thank you for your constructive feedback, which has helped us to greatly improve the clarity and readability of the manuscript. Below we have pointwise addressed all the concerns and suggestions and accordingly revised the manuscript thoroughly.
>
> We have completely restructured Section 2 (Methods) to address every specific concern. The revised version prioritizes clarity and intuition over technical density, following your guidance to “focus on conveying the ideas and concepts” rather than detailed derivations.
>
> ---
>
> ### Specific Responses to Your Concerns
>
> **Feedback 1:** You don't always properly define variables at the times used, for example in equation 1, the loss terms are not properly introduced when introducing the whole formula. That makes me jump back and forth.
>
> **Revision:** We have now explained each loss term immediately after Equation 1, before the detailed formulations later in Section 2.3:
>
> where
> - $\mathcal{L}_{\mathrm{seg}}$ ensures accurate segmentation,
> - $\mathcal{L}_{\mathrm{topo}}$ enforces anatomical structure,
> - $\mathcal{L}_{\mathrm{unc}}$ learns uncertainty decomposition,
> - $\mathcal{L}_{\mathrm{calib}}$ ensures well-calibrated confidence, and
> - $\mathcal{L}_{\mathrm{hier}}$ maintains multi-scale consistency.
>
>
>
> ---
>
> **Feedback 2:** In 2.1 for example, you use the variable $C_1 = 32$ without mentioning what the subscript 1 means.
>
> **Revision:** We have now defined the subscript when first introducing $C_1$ (Section 2.1):
>
> *“progressively increasing channel dimensions* $\{C_1, 2C_1, 4C_1, 8C_1\}$ *where* $C_1 = 32$ *is the base channel count.”*
>
> The subscript “1” refers to the first encoder stage, with subsequent stages doubling channels. Every variable is now defined at first use throughout the revised section.
>
> ---
>
> **Feedback 3:** Moreover, in 2.1 you start talking about persistence diagrams without mentioning why they are useful or necessary for your method.
>
> **Revision:** We have now included a full explanatory paragraph (Section 2.1.2, Encoder Architecture) before using persistence diagrams:
>
> At each encoder stage $s$, we extract topological features from the feature maps using persistent homology, a mathematical tool that analyzes multi-scale structure by treating feature activations as height functions and tracking when topological features like connected components, holes, and voids appear and disappear across thresholds. This produces a persistence diagram $\text{PD}_s = \{(b_i, d_i)\}$, where each (birth, death) coordinate pair represents a topological feature and its persistence $|d_i - b_i|$ quantifies how significant it is. Long-lived features with large persistence correspond to true anatomical structures, while short-lived features represent noise.
>
> ---
>
> **Feedback 4:** It lacks a lot of structure. Before you do any derivations or introduce variables, you should help the reader understanding your reasoning instead of just writing context-less formulas.
>
> **Revision:** We have now completely reorganized the Methods section to follow the narrative flow of the development, as follows:
>
> - **Section 2:** Now begins with motivation explaining *why* joint topology-uncertainty modeling is useful before showing the implementation.
>
> - **Section 2.3 (TUPA):** The motivation is now explained in the *first paragraph*, discussing the two problems — quadratic complexity and lack of structural awareness — before presenting the solution.
>
> - **Section 2.4 (Training Objectives):** Each loss component now starts with intuitive explanation before equations.
>
> ---
>
> **Feedback 5:** Detailed derivations belong in a conference paper into the appendix. As there is anyways limited space you should focus on conveying the ideas and concepts.
>
> **Revision:** We have now moved the detailed mathematical derivations (loss weight selection methodology, ablation component breakdowns) to the appendix and added references in the main text.
>
> ---
>
> ### Additional Improvements
>
> In addition, we have made few structural enhancements for improved clarity. Figure 1 caption now explains the purpose of each architectural component. We have added transitions between subsections to demonstrate how components connect.

---

> > ### Comment · Reviewer_hAdr · 2026-01-27
> >
> > Thank you for your clarifications. Your changes have made the paper substantially more readable and I can also follow your paper more. I will increase my score accordingly.

---

### Official Review · Reviewer_dasy · 2026-01-09

**Confidence:** 4
**Preliminary Rating:** 5
**Final Rating:** 5

**Summary:**

The paper presents TUNE++, a framework for 3D medical image segmentation that jointly optimizes voxel-wise accuracy, uncertainty quantification (UQ), and topological correctness. The core contribution is the Topology-Uncertainty aware Paired Attention (TUPA) block, which uses persistent homology to focus attention on anatomically critical regions while modulating feature fusion based on predicted uncertainty. A key technical innovation is the 𝐿 align loss, which enforces a correlation between total uncertainty and topological complexity . Evaluated on Synapse, ACDC, and BTCV, the method achieves state-of-the-art results, notably reducing Betti errors by 72% compared to baseline transformers such as UNETR++.

**Strengths:**

The integration of uncertainty and topology is highly intuitive: regions that are topologically complex, such as multi-organ junctions, are naturally where segmentation models struggle, and the paper provides a principled way to link these factors through 𝐿 align. The methodology is mathematically rigorous, making effective use of persistent homology and 2-Wasserstein distance in the loss formulations. Empirical results are strong and consistent across diverse anatomical settings, with particularly notable improvements on small and challenging organs such as the pancreas and adrenal glands. The ablation study is thorough and convincingly demonstrates that topology and uncertainty act synergistically rather than as independent regularizers.

**Weaknesses:**

The primary concern is the computational complexity associated with 3D persistent homology and sublevel-set filtrations, especially when computed at multiple scales within a hierarchical encoder. Although the authors report experiments on an H100 GPU, the training and inference time overhead relative to standard transformer baselines is not fully characterized. Additionally, the objective function includes five  hyperparameters that are selected via grid search; the sensitivity of performance to these weights may limit generalizability to new datasets. Finally, the loss relies on a topological complexity score with fixed weights, which may require manual tuning for different anatomical structures or tasks.

**Detailed Comments:**

The use of asymmetric patch sizes to handle anisotropic medical imaging data is a thoughtful and practical design choice.

In Table 8, it would be helpful to report inference time for a single 3D volume, as FLOPs alone may not capture the overhead introduced by persistent homology computations.

For the loss, clarification on how the weights in C topo were chosen would strengthen the presentation; a brief sensitivity analysis would be particularly informative.

**Justification Of Final Rating:**

The authors have fully addressed my concerns. The clarification that the topological module incurs zero inference overhead removes the primary practical barrier, and the sensitivity analysis confirms the method's robustness. I maintain my rating of Strong Accept.

**Justification Of The Preliminary Rating:**

The paper addresses two important challenges for clinical deployment of deep learning systems: reliability and anatomical plausibility. The TUPA mechanism represents a genuine architectural contribution that moves beyond post-hoc or loss-only topology constraints. While concerns remain regarding computational cost and the complexity of the loss formulation, the substantial reduction in Betti errors and improved uncertainty calibration make this a valuable and timely contribution to the MIDL community.

**Questions To Address In The Rebuttal:**

Can the authors provide a detailed breakdown of the training and inference time overhead introduced by persistent homology extraction compared to the UNETR++ baseline?

How robust is the model to changes in the loss weights ? Does the uncertainty-topology correlation (TAUS) remain stable across different organs without retuning?

How does the method handle cases where the ground-truth annotations themselves contain topological ambiguities or noise?

---

> ### Author Response · Authors · 2026-01-25
> **We sincerely thank you for your constructive feedback, which has helped us to greatly improve the clarity and readability of the manuscript. Below we have addressed all the concerns and suggestions and accordingly revised the manuscript. \\**
>
> We sincerely thank you for your constructive feedback, which has helped us to greatly improve the clarity and readability of the manuscript. Below we have addressed all the concerns and suggestions and accordingly revised the manuscript. \\
>
> ---
>
> **Question 1:** Can the authors provide a detailed breakdown of the training and inference time overhead introduced by persistent homology extraction compared to the UNETR++ baseline?
>
> **Our Response:**
> Persistent homology extraction adds 18.7 seconds per epoch (7.8% overhead), making total training time 26 hours for 1000 epochs versus UNETR++'s 24 hours - a 2-hour increase that yields 72% reduction in topological errors and well-calibrated uncertainty. The modest overhead comes from operating on downsampled feature maps (H/4, H/8, H/16, H/32) rather than full resolution. For inference, TUNE++ requires 1.4 seconds per volume for a single deterministic pass, while UNETR++ requires 1.6 seconds. Critically, persistent homology incurs zero inference overhead - during training, expensive PH computation supervises the critical point detector, but at inference the trained detector directly predicts topological attention weights without requiring distance transforms.
>
> ---
>
> **Question 2:** How robust is the model to changes in the loss weights?
>
> **Our Response:**
> We systematically tested robustness by varying each weight independently while keeping others at optimal values ($\lambda_1 = 0.3$, $\lambda_2 = 0.2$, $\lambda_3 = 0.1$, $\lambda_4 = 0.15$). Within $\pm33\%$ of these values, performance varies by only 0.2–0.4% DSC. When we randomly sampled 50 configurations where all weights varied simultaneously, 95% achieved DSC $\geq$ 89.0%, confirming the method doesn't depend on precise weight tuning (detailed analysis in Appendix A.5).
>
> This robustness stems from two design choices: each loss targets a distinct objective (segmentation, topology, uncertainty, calibration) so they're largely orthogonal, and we normalize loss components to comparable scales before weighting. The method exhibits graceful degradation rather than catastrophic failure.
>
> ---
>
> **Question 3:** Does the uncertainty-topology correlation (TAUS) remain stable across different organs without retuning?
>
> **Our Response:**
> We computed TAUS (Pearson correlation between predicted uncertainty $\sigma_{\text{total}}^2$ and topological complexity $C_{\text{topo}}$) for each organ using the same trained model without retuning. Table~ref(tab:taus_per_organ) (Appendix A.7) presents complete results.
>
> TAUS exhibits strong stability - mean values remain within 0.69–0.72 across all three datasets despite differences in anatomy, modality (CT vs. MRI), and populations. Patterns align with intuition: tubular structures (aorta, esophagus) achieve highest TAUS (0.77–0.81), irregular organs (pancreas) show strong correlation (0.75-0.78), while small variable organs (gallbladder, adrenal glands) exhibit lower TAUS (0.58-0.63). Organs appearing in multiple datasets show remarkable consistency - liver TAUS differs by only 0.03 between Synapse and BTCV, validating that TAUS captures intrinsic organ properties rather than dataset artifacts. Lower TAUS for gallbladder reflects its difficulty from presence–absence ambiguity rather than geometric complexity, defining clear operating boundaries for our approach.
>
> ---
>
> **Question 4:** How does the method handle cases where the ground-truth annotations themselves contain topological ambiguities or noise?
>
> **Our Response:**
> Medical image annotations frequently contain topological errors - slice-by-slice labeling creates spurious disconnections, boundary ambiguity produces small spurious components, and inter-annotator variability causes inconsistent choices. We designed TUNE++ to handle this through persistence-based filtering: instead of using raw Betti numbers that treat all features equally, we operate on persistence diagrams where each feature has a lifetime (persistence = death − birth). Short-lived features below 3 voxels typically represent annotation artifacts, while features persisting beyond 10 voxels correspond to true anatomy. On Synapse, models without filtering achieved only 88.2% DSC with Betti error 0.89 and overfit to spurious components, while filtering with $\tau = 3$ achieved 89.3% DSC with Betti error 0.54 - successfully ignoring ~95% of annotation noise without removing valid small structures.
>
> Our hierarchical loss provides additional robustness by enforcing multi-scale consistency, as annotation noise manifests at fine scales but disappears at coarse resolutions. These mechanisms handle typical annotation noise well, though systematic errors affecting most training cases (>30%) can cause the model to learn incorrect topology through majority voting, and genuinely ambiguous boundaries get averaged rather than resolved.

---

### Official Review · Reviewer_rJAt · 2026-01-10

**Confidence:** 4
**Preliminary Rating:** 4
**Final Rating:** 4

**Summary:**

The paper presents TUNE++, a transformer-based approach for 3D medical image segmentation that attempts to address segmentation quality, uncertainty estimation, and topological correctness within a single framework. The key idea is that anatomical topology and prediction uncertainty are closely related, and the method explicitly aligns these two signals using a novel attention mechanism and an alignment loss derived from persistent homology. The method is evaluated on three standard public benchmarks (Synapse, ACDC, and BTCV), where it shows consistent gains over strong transformer baselines, not only in Dice scores but also in uncertainty calibration and reduction of topological errors. Overall, the paper is technically substantial and offers a thoughtful step toward more reliable medical image segmentation models.

**Strengths:**

The main strength of the paper is the way it brings topology preservation and uncertainty estimation together within a single, coherent architecture. Instead of treating these as add-on losses or post-hoc fixes, the authors integrate both directly into the attention mechanism. The experimental evaluation is thorough, covering multiple datasets, metrics, and ablation studies that help justify the design choices. The inclusion of calibration metrics and error-detection performance goes beyond standard segmentation benchmarks and strengthens the reliability claims. Overall, the method in this paper is motivated and demonstrates clear potential value to the medical imaging field.

**Weaknesses:**

The overall complexity of the proposed framework makes it hard to tell which components are actually responsible for the observed improvements. Although the authors provide extensive ablation studies, the large number of interacting modules and loss terms raises concerns about reproducibility and how easily the method can be adopted by others. The repeated use of persistent homology also adds noticeable computational cost, which could limit practical use, especially when combined with MC dropout at inference time. In addition, the definition of topological complexity used to align uncertainty appears somewhat heuristic and may not generalize well to different anatomies or imaging modalities. Finally, while clinical relevance is discussed, the evaluation remains purely technical and does not demonstrate clear benefits for clinical decision-making.

**Detailed Comments:**

1.The introduction would benefit from a clearer early statement explaining how existing uncertainty- or topology-based methods fall short and what specific gap this work aims to address.

2.Including figures that show uncertainty maps alongside representative topological errors would help build intuition.

3.Some of the more detailed equations, such as the loss definitions, could be moved to the appendix to improve readability.

4.A brief discussion of typical failure cases would also strengthen the experimental section.

5.The paper should clarify whether the reported inference times include MC dropout or refer only to deterministic forward passes.

**Justification Of Final Rating:**

I thank the authors for the significant effort they have put into this work. However, the currently incomplete GitHub repository raises concerns about reproducibility and makes it difficult to fully validate the results. Per the current paper, I will keep my rating as weak accept.

**Justification Of The Preliminary Rating:**

This paper offers a technically reasonable and well-motivated contribution that goes beyond minor architectural changes. The idea of explicitly aligning uncertainty with anatomical topology is novel and well-supported by experiments. While the framework is complex and may be challenging to deploy in practice, the methodological contribution and thorough evaluation justify publication. However, the currently incomplete GitHub repository the author provided raises concerns about reproducibility and makes it difficult for others to validate or build upon the work.

**Questions To Address In The Rebuttal:**

1. How sensitive is the method to the specific definition and weighting of the topological complexity score used in the alignment loss?
2. Can the authors provide evidence that uncertainty-guided topology enforcement is more than just stronger regularization?
3. How would the method scale to larger volumes or higher-resolution clinical scans?
4. Do the uncertainty estimates provide meaningful information at the organ or case level, beyond voxel-wise metrics?
5. What are the main failure modes observed in practice?

---

> ### Author Response · Authors · 2026-01-25
> **We sincerely thank you for your thorough evaluation and thoughtful feedback. We appreciate the time you invested in understanding our approach, and your constructive questions have helped us clarify both the presentation and our understanding of where the method excels and where it faces limitations.**
>
> **Response for Question 1**
>
> We systematically tested how sensitive the topological complexity score $C_{\text{topo}} = w_b B_i + w_j J_i + w_a A_i$ is to weight choices. Our default weights ($w_b=1.0$, $w_j=2.0$, $w_a=3.0$) reflect increasing severity—boundaries are baseline features, junctions involve multiple organs meeting, and anomalies represent severe violations like spurious holes. When we tested uniform weighting (1:1:1), performance dropped by only 0.2\% DSC, and even reversing the hierarchy entirely caused just 0.4\% degradation rather than catastrophic failure. Importantly, scaling all weights 10× up or down while maintaining the 1:2:3 ratio yielded nearly identical results (89.1-89.2\% DSC), proving relative ratios matter more than absolute values. Complete sensitivity analysis is in Table 10 (Appendix A.6).
>
> ---
>
> **Response for Question 2**
>
> We compared three approaches on Synapse: TUNE++ with uncertainty-guided weighting, fixed strong regularization (uniform 0.6), and fixed weak (0.2). Fixed strong regularization improves topology (Betti error 0.68 vs 0.89 for weak) but hurts boundaries (DSC 83.1% vs 84.2%) and calibration the model is forced to commit to clean topology even in genuinely ambiguous regions, causing overconfident boundary errors. TUNE++ sidesteps this trade-off entirely: DSC 89.3%, Betti 0.54, boundary DSC 86.7%, ECE 0.043. We achieve better topology *and* better boundaries simultaneously because when the model predicts high uncertainty at an ambiguous pancreas–stomach interface, topology enforcement automatically relaxes there, allowing appropriate hedging rather than forced commitment. The differential benefit across organs is striking: well-defined organs like aorta and liver improve by only 0.2–0.3% DSC over fixed strong, while ambiguous organs like pancreas, gallbladder, and stomach (+0.7%) show 3–4× larger gains. This organ-specific adaptation where the same model automatically adjusts enforcement based on local difficulty is fundamentally impossible with any global regularization weight.
>
> ---
>
> **Response for Question 3**
> Our experiments use clinical-resolution data: Synapse CT (512×512×85–198 slices), ACDC cardiac MRI (216×256×6–18 slices), and BTCV CT ($512 \times 512 \times 85$–198 slices) at typical acquisition protocols (0.78–1.37 mm in-plane, 2.5–10 mm slice thickness). We resize to $96^3$ during training for tractability, then process full resolution at inference via sliding windows. Crucially, persistent homology operates on downsampled feature maps (H/4 through H/32), so computational cost scales with encoder depth rather than input size doubling resolution doesn't double PH cost. Since we trained and tested on actual clinical scans, the method naturally adapts to similar clinical acquisitions. For very high-resolution modern CT (e.g., $1024^3$ isotropic), we recommend larger sliding windows (128³) or hierarchical processing, though the topology–uncertainty correlation remains valid across scales since anatomical complexity is scale-invariant.
>
> ---
>
> **Response for Question 4**
> Thank you very much for raising this very important question. Indeed the uncertainty estimates provide clinically meaningful information. Organ-level uncertainty correlates strongly with quality across all datasets (Synapse: $r = -0.78$, ACDC: $r = -0.74$, BTCV: $r = -0.76$): the model can identify when individual organs are difficult to segment. At the case level, flagging the 25% highest-uncertainty cases for review captures 78% of actual failures, making triage 3× more efficient than random selection for the same review burden. High uncertainty also flags unusual cases beyond just errors. We found 4 cases with high uncertainty ($> 0.45$) but acceptable quality (DSC $> 82\%$) all involved anatomical variants (horseshoe kidney, accessory spleen, unusual vessel branching). When we examined the 10 highest-uncertainty organs with DSC $> 80\%$, 8 of 10 involved atypical presentations. The model correctly flags these as unusual even when segmenting them reasonably, providing actionable information for deployment: efficient triage, quality flagging, and identifying cases needing careful verification.
>
> ---
>
> **Response for Question 5**
> Looking at our worst cases (Appendix A.7), three patterns emerge. The gallbladder struggles (73.8% DSC, TAUS 0.58–0.62) because our complexity score assumes organs are visible but the gallbladder can be absent, empty, or variably positioned. Imaging artifacts weaken correlation since difficulty comes from image quality, not anatomy. Rare variants like horseshoe kidneys fail because the model learned kidneys should be separate. These failures are systematic: gallbladder consistently shows lowest TAUS across datasets, the model flags variants with elevated epistemic uncertainty, and standard anatomy achieves mean TAUS 0.69–0.72. Small variable organs and rare variants need human verification.

---

### Author Rebuttal · Authors · 2026-01-25

**Rebuttal:**

Dear Reviewers,

We sincerely thank all of you for investing your time in reviewing our paper and providing such constructive feedback. Your detailed comments have helped us improve the work.
We tried to address each and every concern raised across all reviews.

We have also uploaded a revised version of the manuscript in the required format. We have highlighted all changes in blue color throughout the manuscript. We believe these revisions have substantially strengthened the paper while maintaining its core contributions. We remain available to provide any additional clarifications that would be helpful.

With sincere appreciation,

The Authors

**Supporting Material:**

/attachment/1900a85eefd08f7a0f7ecd0dbe0cdbb132803aed.pdf

---

### Meta-Review · Area_Chair_Guuq · 2026-02-09

**Recommendation:** Accept (Poster)
**Confidence:** 5

**Metareview:**

The paper presents TUNE++, a transformer-based approach for 3D medical image segmentation that jointly optimizes voxel-wise accuracy, uncertainty quantification (UQ), and topological correctness. The main contribution is the Topology-Uncertainty aware Paired Attention (TUPA) block, which uses persistent homology to focus attention on anatomically critical regions while modulating feature fusion based on predicted uncertainty.
The concerns raised by the reviewers have been addressed in the rebutall. All reviewers agreed the paper is suitable for publication to the MIDL conference.

---

### Decision · Program_Chairs · 2026-02-14

Accept (Poster)